# Apicomplexan mitoribosome from highly fragmented rRNAs to a functional machine

Chaoyue Wang[1,8], Sari Kassem[2,8], Rafael Eduardo Oliveira Rocha[3,8], Pei Sun [4], Tan-Trung Nguyen [3], Joachim Kloehn [2], Xianyong Liu[1], Lorenzo Brusini [2], Alessandro Bonavoglia[2], Sramona Barua[3], Fanny Boissier[3], Mayara Lucia Del Cistia[3], Hongjuan Peng [5], Xinming Tang [6], Fujie Xie[1], Zixuan Wang [1], Oscar Vadas [2], Xun Suo [1]✉, Yaser Hashem [3]✉, Dominique Soldati-Favre [2]✉ & Yonggen Jia [7]✉

The phylum Apicomplexa comprises eukaryotic parasites that cause fatal diseases affecting millions of people and animals worldwide. Their mitochondrial genomes have been significantly reduced, leaving only three protein-coding genes and highly fragmented mitoribosomal rRNAs, raising challenging questions about mitoribosome composition, assembly and structure. Our study reveals how *Toxoplasma gondii* assembles over 40 mt-rRNA fragments using exclusively nuclear-encoded mitoribosomal proteins and three lineage-specific families of RNA-binding proteins. Among these are four proteins from the Apetala2/Ethylene Response Factor (AP2/ERF) family, originally known as transcription factors in plants and Apicomplexa, now repurposed as essential mitoribosome components. Cryo-EM analysis of the mitoribosome structure demonstrates how these AP2 proteins function as RNA binders to maintain mitoribosome integrity. The mitoribosome is also decorated with members of lineage-specific RNA-binding proteins belonging to RAP (RNA-binding domain abundant in Apicomplexa) proteins and HPR (heptatricopeptide repeat) families, highlighting the unique adaptations of these parasites. Solving the molecular puzzle of apicomplexan mitoribosome could inform the development of therapeutic strategies targeting organellar translation.

The superphylum Alveolata diverged approximately 850 million years ago into two main branches: ciliates and myzozoans. The Myzozoa, which includes Apicomplexa, Chromerida, Perkinsozoa, and Dino-flagellates, consists of obligate intracellular parasites and predators that feed through cellular vampirism. These organisms are of significant interest due to their impact on ecological systems and human health, especially given that *Toxoplasma gondii* and *Plasmodium* species cause life-threatening diseases in humans. Most apicomplexans

[1]National Key Laboratory of Veterinary Public Health Security, Key Laboratory of Animal Epidemiology of the Ministry of Agriculture and Rural Affairs, National Animal Protozoa Laboratory & College of Veterinary Medicine, China Agricultural University, Beijing 100193, China. [2]Department of Microbiology and Molecular Medicine, University of Geneva, Geneva, Switzerland. [3]INSERM U1212 Acides nucléiques: Régulations Naturelle et Artificielle (ARNA), Institut Européen de Chimie et Biologie, Université de Bordeaux, Pessac 33607, France. [4]Guangdong Key Laboratory of Animal Conservation and Resource Utilization, Institute of Zoology, Guangdong Academy of Science, Guangzhou, Guangdong Province 510260, China. [5]Department of Pathogen Biology, Guangdong Provincial Key Laboratory of Tropical Diseases Research, School of Public Health; Key Laboratory of Infectious Diseases Research in South China (Ministry of Education), Southern Medical University, 1023-1063 South Shatai Rd, Guangzhou City, Guangdong Province 510515, China. [6]Institute of Animal Science, Chinese Academy of Agricultural Sciences, Beijing, China. [7]Beijing Institute of Tropical Medicine, Beijing Friendship Hospital, Capital Medical University, Beijing 100050, China. [8]These authors contributed equally: Chaoyue Wang, Sari Kassem, Rafael Eduardo Oliveira Rocha. ✉e-mail: suoxun@cau.edu.cn; yaser.hashem@u-bordeaux.fr; Dominique.Soldati-Favre@unige.ch; jarregon@126.com

harbor two endosymbiotic organelles: a mitochondrion acquired when a proto-eukaryote engulfed an α-proteobacterium[1], and an apicoplast, a vestigial plastid resulting from the engulfment of a red alga[2,3]. The mitochondrial genomes of Apicomplexa are characterized by their extreme coding reductionism, leaving only highly fragmented rRNAs, and three protein-coding genes (cob, cox1 and cox3) essential for the mitochondrial electron transport chain (mETC)[4,5]. Mitochondrial ribosomes, or mitoribosomes have also diverged markedly among different species, often displaying a lower RNA-to-protein ratio compared to their bacterial counterparts[6–8], with a few known exceptions, such as green plants[9]. In trypanosomatids, mitoribosomes reveal a significant augmentation of taxon-specific proteins to compensate for the lost segments of mitochondrial rRNA (mt-rRNA)[8,10–13]. Previous studies have reported extensive mt-rRNA fragmentation in apicomplexans[14–17], and characterizations have been limited to individual mitoribosomal proteins (mt-RPs) from both small (mt-SSU) and large mitochondrial ribosomal subunit (mt-LSU)[18–22], but structural data are still lacking. Additionally, the specific composition and evolutionary adaptations that enable apicomplexans to assemble a functional mitoribosome from these highly fragmented rRNAs remain unresolved.

Apicomplexans also feature a diverse array of DNA-binding proteins known as Apicomplexan AP2 (ApiAP2), which bear similarities to plant Apetela2/Ethylene Response Factor (AP2/ERF) transcription factors (TFs). To date, all reported ApiAP2s function as transcription factors, modulating parasite lifecycle transitions by either repressing or activating gene expression alone or in combination[23]. Additionally, the apicomplexan proteome is notably enriched in HPR (heptatricopeptide repeat) proteins[24] and RAP (RNA-binding domain abundant in Apicomplexans) proteins[25]. HPR proteins are involved in the processing and stabilization of mitochondrial RNA[24], whereas the precise functions of RAPs remain largely undefined.

In this work, we identify four AP2 domain-containing proteins (mtAP2-1 to 4) within the mitochondrion of *T. gondii and Plasmodium berghei*, and demonstrate their critical roles in oxidative phosphorylation and survival in *T. gondii*. These mtAP2 proteins, conserved across the Myzozoa lineage, evolve from their ancestral roles in DNA binding and gene regulation to play a crucial part in assembling fragmented rRNAs into functional mitoribosomes. Using mass spectrometry (MS), RNA sequencing (RNA-seq), and cryo-electron microscopy (cryo-EM) on purified *T. gondii* mitoribosomes, we reveal its unique protein-rich composition and the extensive fragmentation of its mt-rRNAs. Alongside mtAP2s, RNA-binding proteins from the RAP and HPR families also contribute to mitoribosome composition and assembly. Our integrative analysis highlights the molecular diversity and distinctive structural features of the mitoribosome, offering new insights into its biogenesis and evolutionary development in Myzozoa.

## Results

### Four AP2 domain-containing proteins in the mitochondrion
The family of ApiAP2s shows a considerable divergence among the Apicomplexa with only few members being lineage-specific[26]. Remarkably, four AP2s are highly conserved in all Myzozoa members that contain mitochondrial genome (mtDNA) (Fig. 1a, b and Supplementary Fig. 1a). In *T. gondii*, one AP2 protein (TGRH88_063820), previously annotated only in the RH-88 strain, has now been added to ToxoDB as TGME49_500252, bringing the total number of AP2 genes to 68 in this species. Three of the four AP2 proteins, AP2IX-6, AP2IV-1 and AP2VIIb-2, were unexpectedly assigned to the mitochondrion by hyperLOPIT dataset in *T. gondii* (Fig. 1a)[27]. We successfully tagged three of them (AP2IX-6, AP2IV-1 and AP2VIIb-2) at their endogenous locus and demonstrated their mitochondrial localization by indirect immunofloresence assay (IFA) (Fig. 1c). However, the fourth AP2 protein, AP2XII-10 was refractory to epitope tagging at the C-terminus and hence its mitochondrial localization was assessed by expressing a second copy (Fig. 1c). Similarly, the homologous of AP2IX-6, AP2IV-1 and AP2VIIb-2 in *P. berghei* were also localized to the mitochondrion (Supplementary Fig. 1b–d). To further support that the AP2 proteins are inside the mitochondrion, we employed the split-GFP complementation technique[28]. The plasmid encoding GFP$_{1-10}$ fused to the mitochondrial matrix targeting sequence (MTS) of HSP60 (MTS-HSP60-GFP$_{1-10}$) was introduced into 7xGFP$_{11}$ tagged AP2 parasites and a strong mitochondrion-specific green fluorescence signal colocalized with HSP70 (Supplementary Fig. 2a, b), thus confirming their localization in the mitochondrial matrix. Consequently, we designated them as mitochondrial AP2 domain-containing proteins 1 to 4 (mtAP2-1 to mtAP2-4) (Fig. 1b). Their unusually fast migration on gels, relative to their predicted molecular weights (Fig. 1a, b, d), suggested the presence of long N-terminal mitochondrial targeting signal (MTS)[29].

Three of the four mtAP2s were previously identified as fitness-conferring genes in a CRISPR/Cas9 genome screening analysis in *T. gondii* (Fig. 1a)[30]. To validate this and to assess their functional roles, we employed Tet-inducible or U1 snRNP-mediated knockdown systems to downregulate their expression upon anhydrotetracycline (ATc) or rapamycin treatment, respectively[31,32]. Consequently, we generated conditional knockdown strains for all four mtAP2s using the Tet-inducible system, designated mtAP2-1-Ty iKD, mtAP2-2-Ty iKD, mtAP2-3-Ty iKD and mtAP2-4 iKD in the text and figures. Additionally, we also established knockdown strains for mtAP2-1 and mtAP2-3 via U1 snRNP-mediated system, referred to as mtAP2-1-Ty-U1 and mtAP2-3-Ty-U1 in the text and figures. Knockdown was successfully achieved, and the mtAP2-(1 to 3) were not detectable after 2 days of treatment (Fig. 1e, f and Supplementary Fig. 2c, d). Plaque assays and intracellular growth quantification revealed significant replication defects upon mtAP2 depletion (Fig. 1g–i and Supplementary Fig. 2e–g). Notably, these defects were reversible upon the withdrawal of ATc after 3 days of treatment (Supplementary Fig. 2h). Since small plaques were still observed upon mtAP2s depletion, a knockout strain (ΔmtAP2-2) was generated (Supplementary Fig. 2i, j) that exhibited a significant growth defect. This defect was completely restored by complementing wild-type mtAP2-2 (Fig. 1j, k and Supplementary Fig. 2k, l). Furthermore, mice infected with the parental virulent RH strain or the mtAP2-2-complemented strain could not survive, whereas those infected with the ΔmtAP2-2 strain survived and mounted a long-term immune response against a challenge with RH tachyzoites (Fig. 1l). Thus, mtAP2s are mitochondrial matrix proteins essential for parasite replication and virulence.

### The mtAP2s do not function as transcription factors
To investigate if the four mtAP2s function as transcription factors like other ApiAP2 proteins, we conducted RNA-seq analysis following the depletion of mtAP2s. Contrary to expectations, none of the four mtAP2s significantly affected global or mitochondrial transcription (Fig. 2a and Supplementary Data 2). Specifically, the transcription levels of the only three mtDNA-encoded protein genes (cox1, cox3, and cob) remained unchanged (Fig. 2a and Supplementary Data 2). Similar results were observed upon mtAP2-2 deletion (Supplementary Fig. 3a, left panel and Supplementary Data 2), confirming that these four mtAP2s do not serve as transcription factors. Contrastingly, depletion of mtAP2s led to a significant reduction in the relative abundance of subunits within respiratory chain Complexes III and IV, while subunits of Complexes II and V remained unaffected (Fig. 2b and Supplementary Data 3). Consistent results were obtained upon deletion of mtAP2-2 (Supplementary Fig. 3a, right panel and Supplementary Data 3). This finding was validated by tagging individual subunits of the *T. gondii* mETC complexes. Specifically, the depletion of the four mtAP2s led to a significant reduction in the protein levels of QCR12 (Complex III) and ApiCox26 (Complex IV), while the levels of subunits in Complexes II (SDHB) and V (F1γ) remained largely unchanged (Supplementary

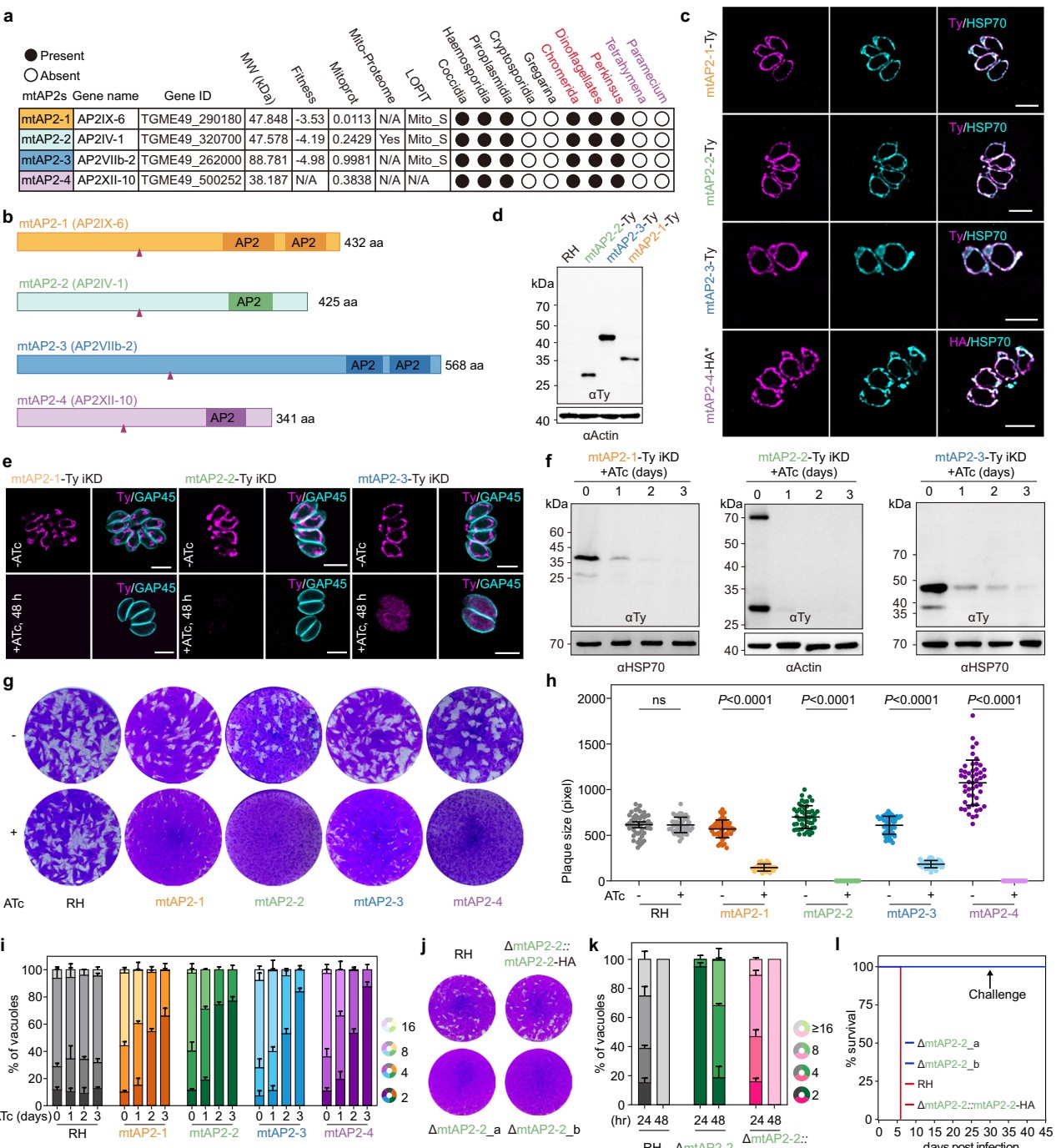

**Fig. 1 | Four conserved AP2 domain-containing proteins in Alveolata (Myzozoa) are fitness-conferring and targeted to the mitochondrion of *T. gondii*.**
**a** Summary of the four mtAP2s in *T. gondii* and their conservation in Alveolates. Fitness score based on the genome-wide CRISPR fitness screen[30], mitochondrial targeting peptide predictions using MitoProt II, mitochondrial proteome based on the BirA* and APEX[73], and localization information based on hyperLOPIT data[27]. Conservation in Alveolates is presented with a filled circle, absence with an empty one. **b** Graphic representation of mtAP2s and their predicted cleavage sites indicated by arrowheads. **c** Immunofluorescence assays (IFAs) of mtAP2s (anti-Ty or anti-HA, magenta) and the mitochondrial marker HSP70 (cyan). Three independent experiments. Scale bar, 5 μm. **d** Immunoblot of endogenous C-terminally 6×Ty tagged mtAP2-(1 to 3). Anti-Ty antibodies were used to detect three mtAP2 proteins and anti-actin antibodies were used as a loading control. Three independent experiments. **e** IFAs of mtAP2-(1 to 3)-Ty iKD parasites treated with ATc or vehicle

for 48 h. Anti-Ty (magenta) and anti-GAP45 (cyan). Three independent experiments. Scale bar, 5 μm. **f** Immunoblots of mtAP2-(1 to 3)-Ty parasites treated with ATc or vehicle for 1-3 days. Anti-Ty (mtAP2s) and anti-actin or HSP70 antibodies (loading control). Three independent experiments. **g** Plaque formation by mtAP2-(1 to 4)-Ty iKD parasites growing on HFF monolayers for 7 days ± ATc.
**h** Quantification of plaques, mean ± SD (n = 3 independent biological replicates). Each mtAP2-regulatable parasite line treated with ATc was individually analyzed for statistical significance using an unpaired two-tailed Student's *t*-test (mtAP2s versus RH), ns: *P* > 0.05. **i**, Intracellular replication of parental line (RH) and mtAP2-(1 to 4)-Ty iKD lines after treatment with ATc or vehicle. Data represented as mean ± SD (n = 3 independent biological replicates). **j, k** Plaque formation (**j**) and intracellular replication (**k**) of RH, ΔmtAP2-2-(a and b) and ΔmtAP2-2::mtAP2-2-HA lines. Data (**k**) represented as mean ± SD (*n* = 2 independent biological replicates). **l** Survival curves of infected mice. Source data are provided as a Source Data file.

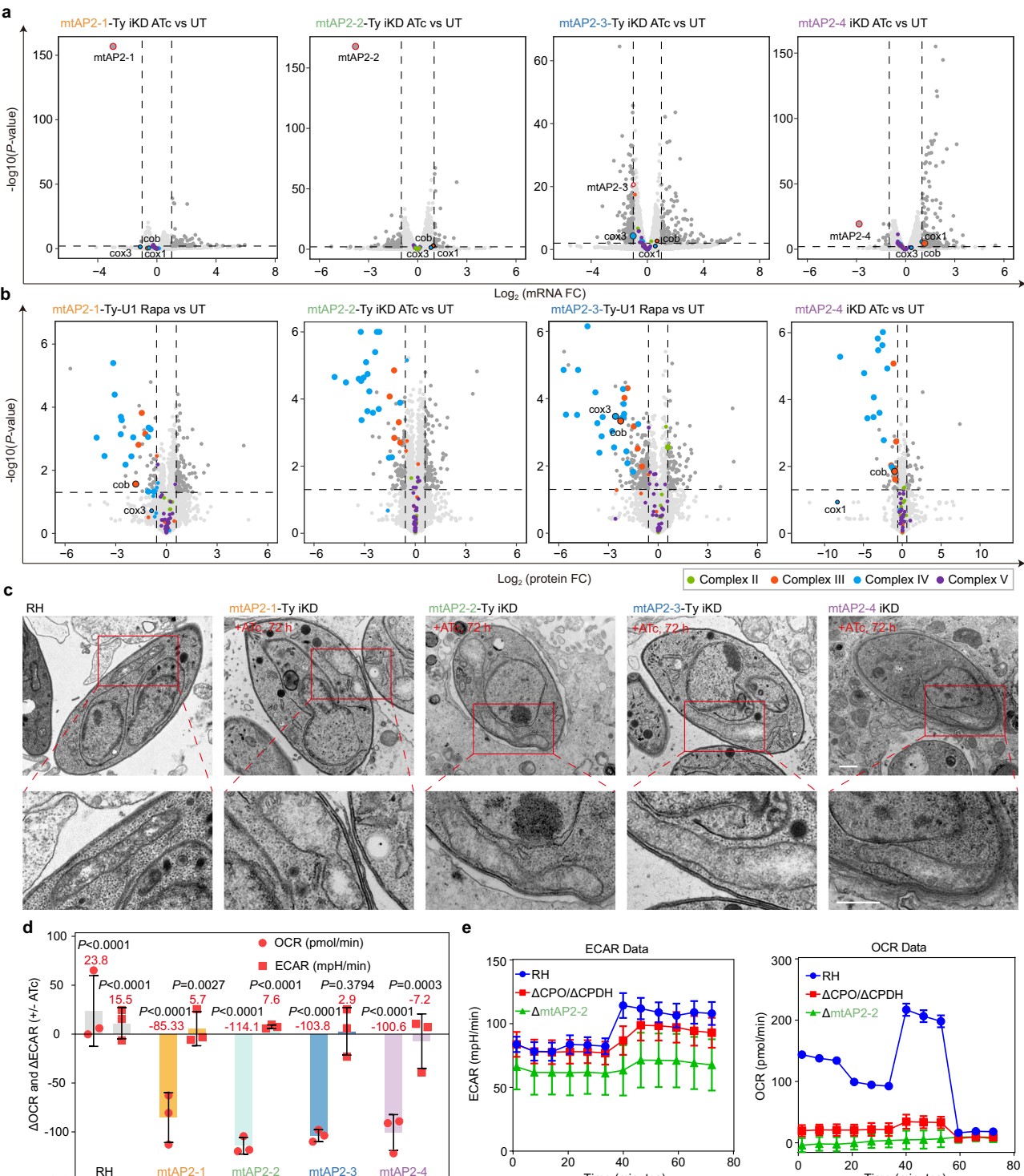

**Fig. 2 | The mtAP2s are required for the normal morphology and function of the mitochondrion in *T. gondii*. a**, **b** Volcano plots highlighting the differential expression genes (DEGs) (**a**) and the differential expression proteins (**b**) of ATc versus vehicle or rapamycin versus vehicle-treated in four mtAP2 knockdown lines. The cutoff for DEGs was a p-value < 0.01 and ± 2-fold change, while the cutoff for differential protein expression was a p-value < 0.05 and ±1.5-fold. X-axis shows log2 fold change, Y-axis shows -log10(*P*-value). **c** Electron micrographs of wild-type line and four mtAP2s knockdown lines for 72 h ± ATc. Insets (red) show representative mitochondrion. Scale bar, 500 nm. **d**, **e** Extracellular flux (Seahorse) analysis of mtAP2s knockdown lines and ΔmtAP2-1. Delta values of the basal oxygen consumption rate (OCR) and the extracellular acidification rate (ECAR) in response to downregulation of four mtAP2s are shown (p-values, top, comparing the -ATc and +ATc condition and the numerical delta values are provided (red), bottom). Bars

represent the means of three independent experiments. Dots indicate the means of these experiments, each based on >18 technical replicates. Error bars represent mean ± SD. Statistical significance was tested using a one-way ANOVA, followed by Bonferroni test for multiple comparisons (**d**). ECAR and OCR traces of a mitochondrial stress test comparing RH, ΔmtAP2-1 and a mutant defective in heme synthesis (Δ coproporphyrinogen III oxidase, CPO/ Δ coproporphyrinogen III dehydrogenase, CPDH)[33]. Oligomycin (2.5 μM), carbonyl cyanide-p-trifluoromethoxyphenylhydrazone (FCCP, 2 μM), and antimycin A (0.5 μM) were injected after the 3rd, 6th, and 9th data points, respectively. The displayed traces are from a single representative experiment performed three times independently. Data points show the means and error bars indicate mean ± SD of >12 technical replicates (**e**). Source data are provided as a Source Data file.

Fig. 3c–e). Using transmission electron microscopy (TEM), we observed abnormalities in mitochondrial morphology following mtAP2 depletion, including swollen and twisted mitochondria with a reduced number of cristae (Fig. 2c and Supplementary Fig. 3b). To directly evaluate the impact of the four mtAP2s on oxidative phosphorylation (OXPHOS), extracellular flux analyses (Seahorse, Agilent) were performed. The results demonstrated a dramatic and significant decrease in the oxygen consumption rates (OCR) of extracellular parasites following mtAP2s depletion, while marginal or no differences were observed in the extracellular acidification rates (ECAR), a measure for glycolysis (Fig. 2d). The observed defect in mitochondrial respiration was comparable to that of a mutant defective in heme synthesis (Δ coproporphyrinogen (CPO) III oxidase/ ΔCPO III dehydrogenase)[33], which was included as a reference (Fig. 2e). To determine the significance of AP2 domains for the function of the four mtAP2 proteins, mutants lacking the AP2 domains were subjected to a functional complementation assay in parasites. The expression of a second copy of wild-type (WT) mtAP2 successfully rescued the growth defect observed in the knockdown strain. In contrast, mtAP2 mutants lacking the AP2 domains, as well as an HA-tagged mtAP2-4 at its C-terminus (mtAP2-4-HA), were unable to rescue the growth defect (Supplementary Fig. 3f). The C-terminal sequences following the AP2 domain in mtAP2-1 and mtAP2-2 were also found to be essential (Supplementary Fig. 3f). Interestingly, the AP2 domain of *T. gondii* could be successfully replaced with an AP2 domain from another apicomplexan, the chicken parasite *Eimeria maxima*, resulting in the restoration of mtAP2-2 function and parasite growth and demonstrating a conserved role for AP2 domains across the Apicomplexa (Supplementary Fig. 3f). Thus, mtAP2 proteins are crucial for maintaining respiratory Complexes III and IV and functional mitochondrion.

## The four mtAP2s associate with the mitoribosome and are required to maintain its integrity

The depletion of mtAP2s selectively impacts Complexes III and IV, both of which harbor mitochondrial-encoded subunits, hinting at a potential involvement of mtAP2s in either RNA processing or translation, which affects the production of cob, cox1, and cox3. To explore these possibilities, we conducted immunoprecipitations (IPs) followed by liquid chromatography-tandem mass spectrometry (LC-MS/MS) to identify potential partners of the four mtAP2s (Fig. 3a). Remarkably, small mitochondrial ribosomal subunit (mt-SSU) proteins were exclusively enriched with mtAP2-1, while large mitochondrial ribosomal subunit (mt-LSU) proteins were associated solely with mtAP2-2 and mtAP2-3 (Fig. 3b, upper panels and Supplementary Data 4). Additionally, mtAP2-4 was found to interact with mtAP2-2 and mtAP2-3, but not with mtAP2-1 (Fig. 3b, upper panels), indicating that mtAP2-1 is specifically linked to the mt-SSU, whereas mtAP2-(2 to 4) are associated with the mt-LSU. To validate these findings, we generated parasite strains with C-terminal FLAG-tagged universal mt-RPs at their endogenous loci for both LSU and SSU, confirming their mitochondrial localization (Supplementary Fig. 4a). IPs via uS5m revealed enrichment of mt-SSU and the mtAP2-1, while uL4m and bL27m IPs were enriched with mt-LSU in addition to the mtAP2-(2 to 4) (Supplementary Fig. 4b and Supplementary Data 5). The specific interactions were further confirmed by reverse IPs wherein mtAP2-1 interacts with bS21m and uS5m (Supplementary Fig. 4c, d), while mtAP2-2 and mtAP2-3 interact with uL4m and bL27m (Supplementary Fig. 4e). Since magnesium ion (Mg$^{2+}$) is essential for the association of the large and small subunits[34], the mtAP2s-IP experiments were repeated in MgCl$_2$ supplemented buffer (Fig. 3b, lower panels). Strikingly, all four mtAP2s were enriched with both the small and the large mitoribosomal proteins (Fig. 3b, lower panels and Supplementary Data 4), indicating that the four mtAP2s are bona fide mitoribosomal proteins. Further support was provided by depleting uL4m (Supplementary Fig. 4g, h), which

reproduced the growth defect (Supplementary Fig. 4i), alterations in mitochondrial morphology (Supplementary Fig. 4j), and compromised integrity of Complexes III and IV (Supplementary Fig. 4k) observed upon mtAP2s depletion (Fig. 2a–c and Supplementary Fig. 3a, b).

When comparing mtAP2s IPs in the absence (- MgCl$_2$) and presence (+ MgCl$_2$) of magnesium, we differentiated lineage-specific mitoribosomal proteins belonging to mt-LSU or mt-SSU. Of the 109 identified *Toxoplasma* mitoribosomal proteins, 45 were linked to the mt-SSU and 64 to the mt-LSU (Fig. 3c). Nearly half of the identified mitoribosomal proteins were novel and specific to Myzozoa (Fig. 3c). Remarkably, members of two lineage-specific families of RNA-binding proteins, RAPs and HPRs were also found among the mitoribosomal proteins (Fig. 3c).

We next performed RNA immunoprecipitations (RIPs) followed by next-generation sequencing (NGS) to determine the rRNA composition of the *Toxoplasma* mitoribosome (Supplementary Fig. 5a). In total, we identified 43 small RNAs (sRNAs) ranging from 21 to 280 nucleotides in length, comprising 17 SSU-rRNA and 26 LSU-rRNA sequences. Among these, 33 were previously reported[17] and we discovered 7 new ones, LSUC, RNA35 and RNA37-RNA41 (Supplementary Fig. 5b–e and Supplementary Data 7). Intriguingly, SSUF, RNA13 and RNA15 were detected in both LSU and SSU fractions (Supplementary Fig. 5b, c).

To test whether mtAP2s are critical for mitoribosome assembly, *T. gondii* complexes were fractionated on the sucrose gradient (Fig. 3d). Expectedly, mtAP2-3, along with uL4m and mitoribosomal rRNA large (LSUA), formed a complex corresponding to the LSU subunit. The mitoribosomal rRNA small (SSUA) was mainly enriched in a smaller complex corresponding to the SSU subunit (Fig. 3e). Notably, both SSUA and LSUA were detected in a higher molecular weight complex corresponding to the mt-monosome (Fig. 3e).

To determine the impact of mtAP2s on the integrity of the mitoribosome, we performed sucrose fractionation post mtAP2s depletion revealing distinct effects for each mtAP2 (Fig. 3f-i). Depletion of mtAP2-1 affected the SSU (Fig. 3f), while depletion of mtAP2-2 impacted the LSU, resulting in the complete disappearance of both uL4m and bL27m from mitoribosomal fractions (Fig. 3g). Depletion of mtAP2-3 affects only bL27m, while the levels of uL4m remained unchanged (Fig. 3h). Similar effects were observed with the down-regulation of mtAP2-4, resembling those seen after depleting mtAP2-3 (Fig. 3i). Additionally, steady-state levels of uL4m, bL27m, and uS5m, as well as other individual mitoribosomal proteins uL3m, bL21m and bS21m, were assessed in total lysates after mtAP2s depletion (Supplementary Fig. 6a–g). Collectively, four mtAP2 proteins are core mitoribosomal subunits, required for the integrity of the mitoribosome.

## Cryo-EM structure of *Toxoplasma gondii* mitoribosome

To gain deeper insight into the roles of the four mtAP2s at an atomic level, we affinity-purified the LSU and mitoribosome[35] from large-scale cultured *T. gondii* tachyzoites using a uL4m-FLAG tagged subunit as bait. The samples were fractionated by sucrose gradient, and the fractions corresponding to mt-LSU and monosome complexes were combined and their cryo-EM structures resolved (Supplementary Fig. 6h, i). The LSU was resolved to 2.89 Å (Supplementary Fig. 7a–d and Supplementary Table 1), and the entire mitoribosome was locally refined to three bodies, the SSU head, the SSU body and the LSU, at resolutions of 3.6 Å, 3.29 Å and 3.11 Å, respectively (Supplementary Fig. 7e–j and Supplementary Table 1). The first striking feature is the relative size of the SSU compared to the LSU (Fig. 4a–d), as the former appears substantially larger with an elongated oval-shaped head (Fig. 4e, f). These complexes reveal extreme species-specific features and confirm the surprising number of additional ribosomal proteins, most of which are Myzozoa-specific (Fig. 4a–f, Supplementary Fig. 8 and Supplementary Data 7). The recruitment of these additional

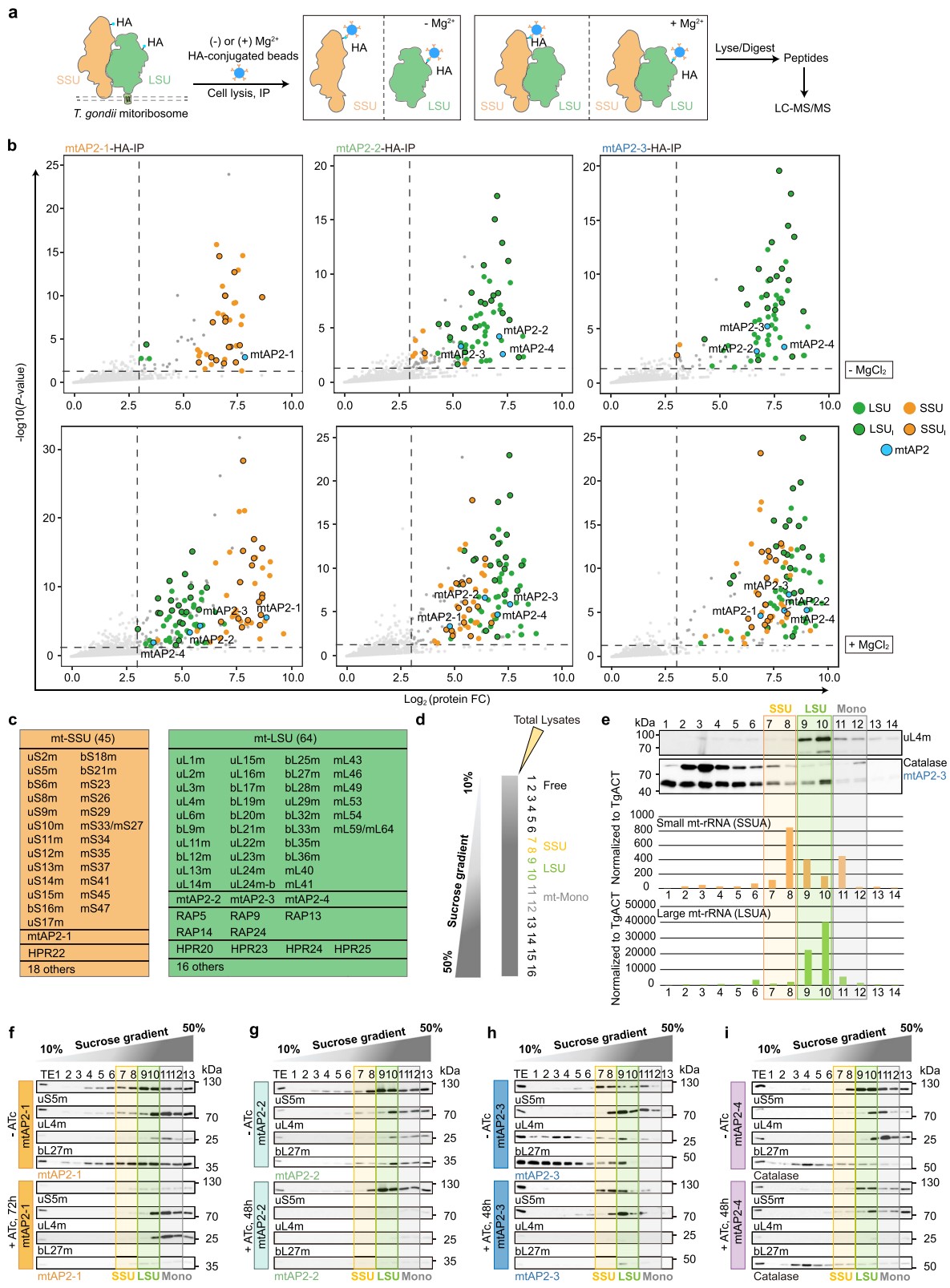

proteins is achieved while conserving a sizable rRNA core in both subunits (Fig. 4g, h), thereby making it larger than the bacterial ribosome (Fig. 4i, j).

Using a semi-automated workflow, we have derived an atomic model for both ribosomal subunits (Fig. 5a–d and Supplementary Table 1). The SSU harbors two particularly protein-rich regions, on the top of the head and the bottom (foot) of the body. On the top of the

head, a cluster of mainly three large proteins (mS29, mS139 and mS141) form a tight domain (Fig. 5a, c) around two short unassigned leftover rRNA fragments (ulr22 and ulr23) (Fig. 5e and Supplementary Fig. 10a). The foot domain is formed by a cluster of mainly two large proteins (mS138 and mS142) (Fig. 5a, c) wrapped around two short rRNAs, one of which was assigned to RNA13 (Fig. 5e and Supplementary Fig. 10a). The SSU presents an extended platform and very short helix 44 (h44)

**Fig. 3 | The four mtAP2s are associated with the mitoribosomes. a** Schematics for the mtAP2-(1 to 3)-HA immunoprecipitations ± MgCl₂. **b** Volcano plots showing the enriched proteins via mtAP2-(1 to 3)-HA immunoprecipitations ± MgCl₂, detected by mass spectrometry. Homologs of universal conserved mtRPs (LSU and SSU) and lineage-specific mtRPs (LSUₗ and SSUₗ) are represented by colored dots. The cutoff for enriched proteins was a p-value < 0.05 and Log2FC ≥ 3. X-axis shows log2 fold change, Y-axis shows -log10(P-value). **c** Schematic representation and summary table of mtRPs based on the differential protein enrichment from various mtAP2s-IPs ± MgCl₂. **d** A schematic representation of the mitoribosome profiling assay. **e** Distribution of mitoribosomal rRNAs after sucrose gradient sedimentation of *T. gondii* cellular extracts expressing uL4m-FLAG and mtAP2-3-Ty, supplemented with RNase inhibitors ( + RNasIn) were analyzed. Fourteen fractions were collected, and mt-rRNA content was quantified. SSUA represented the small subunit, and LSUA represented the large subunit. Results were normalized to TgACT levels. Western blot analysis (Top) is included. Two independent experiments. **f–i** The distribution of uL4m, uS5m, bL27m in the mtAP2-(1 to 4) iKD parasites in the presence or absence of ATc treatment on a 10–50% sucrose gradient fractions. No corresponding western blot for mtAP2-4 is available due to unsuccessful attempts to tag this protein. Immunoblot for catalase instead. Two independent experiments. Source data are provided as a Source Data file.

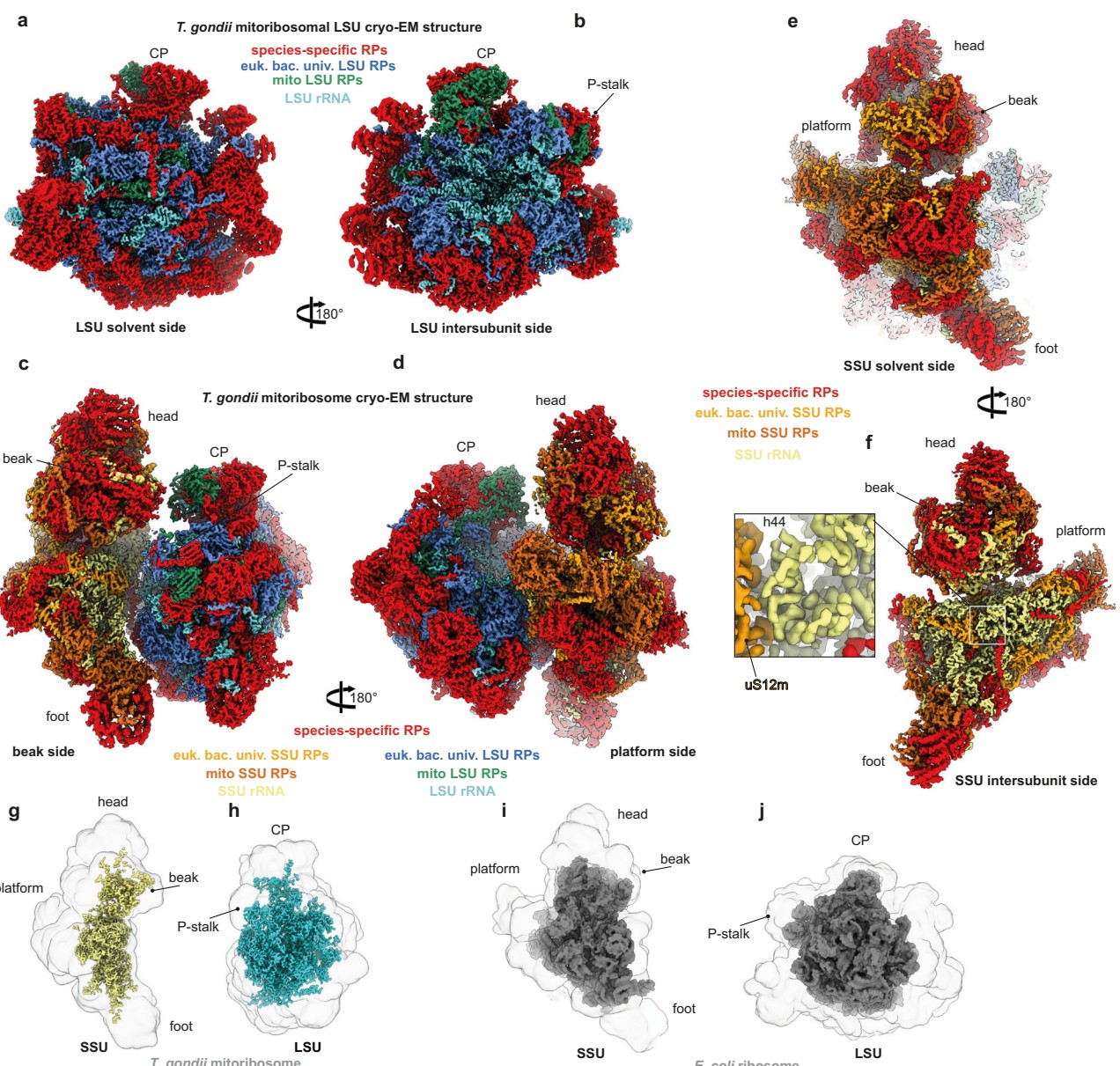

**Fig. 4 | The overall structure of *T. gondii* mitochondrial ribosome. a, b** Solvent side (**a**) and intersubunit side (**b**) views of the cryo-EM reconstruction of the *Toxoplasma* mitoribosomal large subunit from the LSU sample. The LSU components are colored according to their composition and conservation. **c–e** Structure of the full mitoribosome, from the monosome sample, seen from the beak side (**c**), the platform side (**d**) and the small ribosomal subunit (SSU) side (**e**). The LSU and SSU components are colored according to their composition and conservation.

**f** Focuses on the intersubunit face of the SSU. The helix (h44) contains the A site (decoding center) that interacts with the uS12m r-protein. **g, h** 3D representation of the rRNA proportion in the *T. gondii* mitoribosome, yellow for the SSU and cyan for the LSU. The r-proteins are shown as gray silhouettes. **i, j** Sizes of the SSU and LSU of the *T. gondii* mitoribosome comparatively to the *E. coli* bacterial ribosome. CP: central protuberance.

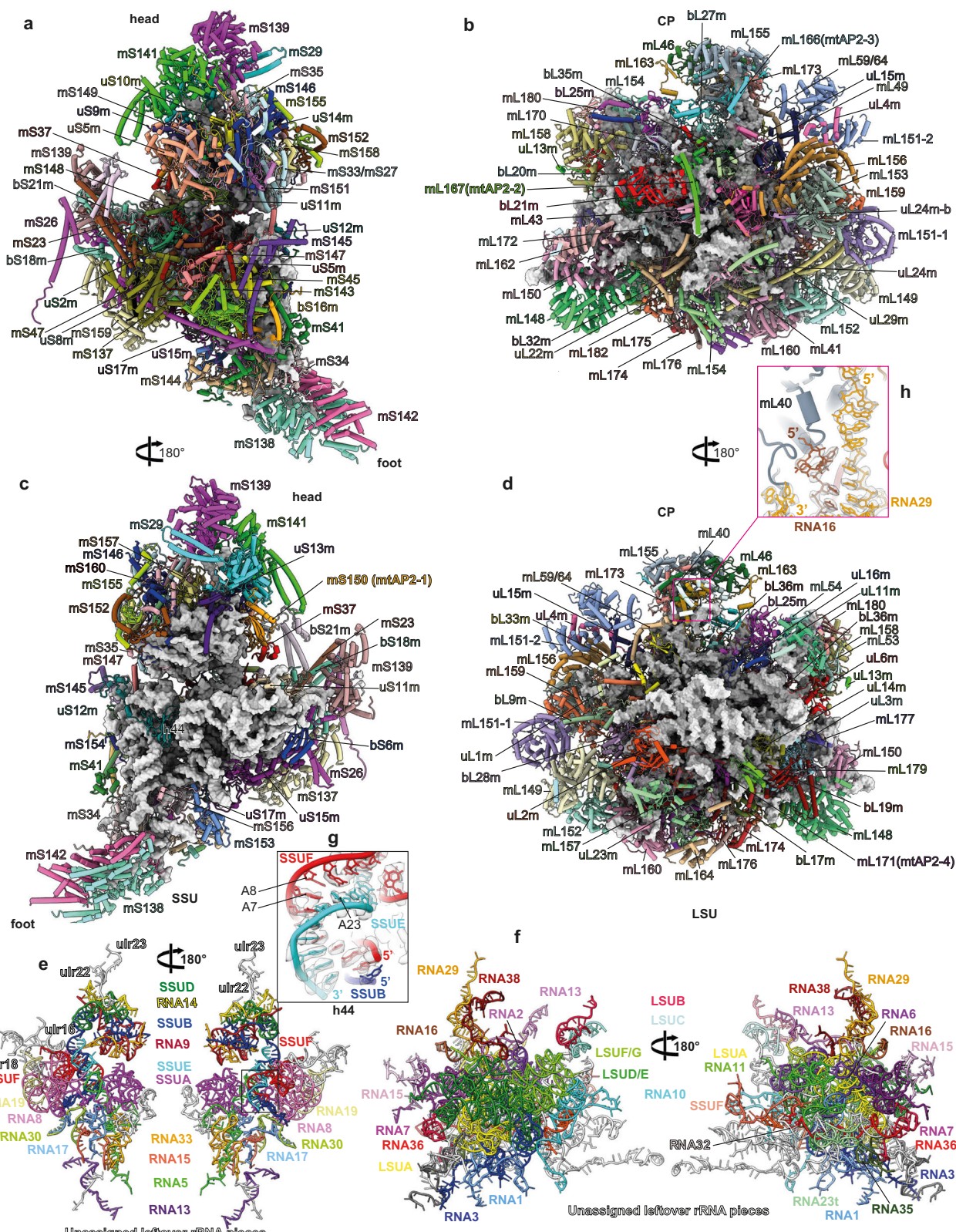

**Fig. 5 | Atomic model of the *T. gondii* mitoribosome. a–d** The individually colored mitoribosomal proteins are shown in cartoon and the rRNAs in gray surface. Atomic model of the mitoribosomal SSU and LSU, respectively, seen from the solvent side (**a**, **b**) and the intersubunit side (**c**, **d**). **e**, **f** Arrangement of mitoribosomal rRNA fragments in the SSU (**e**) and LSU (**f**). **g** Zoomed-in view of the h44 region forming the A site by SSUE, SSUF and fraction of SSUB. **h** Zoomed-in view of the only rRNA fragment passing through the CP showing that mL40 stabilizes the 5′end of RNA29 extending towards the top of CP. In both (**g**) and (**h**), the cryo-EM density contours are shown around the rRNAs. CP: central protuberance.

(Fig. 4f) that harbors the A-site (decoding center), which is universally conserved across all ribosomes and species. The universally conserved three-adenines bulge forming the A-site is formed mainly by SSUE (harboring A23, corresponding to A1408 in bacteria) and SSUF rRNA fragments (A7 and A8, corresponding to A1492 and A1493 in bacteria, respectively) (Fig. 5g), with minor participation from SSUB 5′ extremity.

Interestingly, the universally conserved uS3m at the entrance of the mRNA binding channel is absent in the *T. gondii* mitoribosome (Supplementary Fig. 9a, b) and is instead replaced by the C-terminal and N-terminal tails of mS151 and mS148. Moreover, the entrance of the mRNA channel is surrounded by the mitochondria conserved mS33/mS27, the C-terminal helix and an unstructured loop from the species-specific mS147 and the universally conserved uS5m (Supplementary Fig. 9b). Also, uS5m is substantially enlarged in this mitoribosome, as it has acquired a large SSU-head domain and N-terminal and C-terminal extensions on the SSU body (Supplementary Fig. 9c, d). Finally, one must highlight the absence of the conserved uS7m at the exit of the mRNA channel in *T. gondii*, a position that seems to partially overlap with mtAP2-1 (Supplementary Fig. 9a, b).

As observed in trypanosomatids[8,36] and ciliates[37], the central protuberance (CP) of the *T. gondii* mitoribosome lacks 5S rRNA (Supplementary Fig. 9e). The CP consists of the conserved mitoribosomal proteins mL40, mL46, mL59/64, which are found in other species, as well as Myzozoa-specific mtRPs mL155, mL163, mL173, and mtAP2-3 (Supplementary Fig. 9f). This protein-rich CP is bound on 4 rRNA fragments (RNA13, RNA16, RNA29 and RNA38), recapitulating part of the 23 rRNA between residues 2234 and 2405 in bacteria (Supplementary Fig. 9g, h). Despite the lack of a 5S rRNA, the 5′ of RNA29 extends out and towards the top of the CP and is tightly embedded in several ribosomal proteins, mainly mL40 (Fig. 5h and Supplementary Fig. 9h). Notably, uL24m possesses an ortholog, uL24m-b, that binds in its vicinity on a different site (Supplementary Fig. 9i, j). Similarly, mL151 (RAP9) exists in two copies identical in sequence and conformation, yet bind at two different sites, relatively distant from each other (Supplementary Fig. 9i, k).

Most strikingly, the rRNAs in both subunits are fragmented into numerous pieces (25 fragments in the LSU and 15 in the SSU that are assigned, and 23 fragments that are not assigned across both subunits) (Fig. 5e, f and Supplementary Data 7), thus nearly three folds more fragmentation than observed in the mitoribosome from the microscopic algae *C. reinhardtii*[38]. Although most of these rRNA fragments can be mapped onto the bacterial rRNA as previously reported[14], some fragments are unique and do not present counterparts in bacteria, and vice-versa (Supplementary Fig. 10). Moreover, despite the sufficient definition and resolution in most regions, especially on the LSU, some fragments were not assigned to any sequences characterized by our RIP-seq analysis. We chose to represent unassigned fragments as poly-U fragments in our deposited structures and to label them with the generic name "ulr". There is a significant reduction in the total rRNA mass compared to the bacterial ribosome (Figs. 4g, h and 5e, f and Supplementary Fig. 10). On the SSU rRNA, SSUB is the center piece that goes from the body to the head (Fig. 5e). In the body, it interacts with SSUE and SSUF to form the A-site (Fig. 5g). It is also engaged in base-pairing with RNA5, SSUA and RNA8 (Supplementary Fig. 10a). By base-paring with SSUE, it forms the neck region and runs to the head where it base-pairs with RNA9 forming the main portion of the beak alongside SSUD.

As for the LSU rRNA fragments, the peptidyl transferase center (PTC) region is nearly strictly conserved compared to any ribosome and is recapitulated by LSUD/E the center piece of the LSU, engaged in base-pairing interactions with RNA3, LSUF/G, RNA13, RNA1 and SSUF (Fig. 5f and Supplementary Fig. 10b). Interestingly, SSUF is an RNA fragment that found in the SSU as part of the A-site (Fig. 5e, f) and as

part of domain I and forms a stem-loop structure that does not feature in the bacterial LSU (Supplementary Fig. 10) but in the SSU. While RNA domains II and III are largely reduced, with a dozen helices, compared to the bacterial counterpart, domains IV and V are mostly conserved where the rRNA fragments recapitulate most of the conserved helices, and Domains I and VI appear highly fragmented but most of the key features are retained (Supplementary Fig. 10b).

Given the high fragmentation, some RNAs have evolved to act as "band aids" gluing fragments in close proximity (ulr1 and RNA14), which are otherwise distant from a sequence stand view (Supplementary Fig. 10b, dashed red lines). In both LSU and SSU, some rRNA fragments are connected to the rest of the structure through ribosomal protein interactions, such as fragments LSUB and LSUC that recapitulate the P-stalk rRNA (Fig. 5f and Supplementary Fig. 10b). Finally, the L1-stalk rRNA, invariably formed by helices 76, 77 and 78 (Supplementary Fig. 10b) could not be mapped to any rRNA fragments in our structure, as the position of L1-stalk remains completely unoccupied by any rRNA or protein structure.

### The four mtAP2s assemble rRNA fragments in the mitoribosome

As anticipated, mtAP2-1 on the SSU is located at the head-to-body junction near the platform (Fig. 6a). The remaining three mtAP2s (mtAP2-2 to mtAP2-4) are positioned on the LSU: one at the central protuberance (Fig. 6c), another above the nascent peptide channel exit on the solvent side (Fig. 6e), and the third below the PTC on the inter-subunit side (Fig. 6g). All these mtAP2s bind at the 3′ and 5′ extremities of rRNA fragments, suggesting their crucial role in assembling peripheral rRNA fragments during mitoribosome maturation.

Specifically, mtAP2-1 interacts mainly with fragments SSUB and SSUD with its first AP2 domain binding directly to the 3′ end of SSUD and interacting minorly with SSUB, mostly via arginine residues with the RNA backbone (Fig. 6a, b). The mtAP2-3 is located at the CP and directly interacts with the 3′ ends of RNA13 and RNA29 and the 5′ end of RNA38 (Fig. 6c, d). These interactions are mediated by arginine residues engaging with the RNA backbone rather than forming pseudo-pairing interactions, sometimes involving stacking interactions between the base and aromatic side chains (Fig. 6d). This positioning suggests mtAP2-3 plays a crucial role in maintaining the CP, which lacks 5S rRNA and where the rRNA is fragmented (Supplementary Fig. 9f, h). mtAP2-2 shows extensive interactions with RNA11 and the 3′ and 5′ extremities of LSUA (Fig. 6e, f). These interactions primarily involve arginine residues with the RNA backbone and some stacking interactions (Fig. 6f). Finally, mtAP2-4 is located at the inter-subunit face, below the PTC, at the interface with the SSU (Fig. 6g, h). This positioning explains why attempts to tag its C-terminal failed. The main interaction interface involves the AP2 domain engaging with LSUA and the 3′ end of RNA3 (Fig. 6h), using several arginine residues in addition to some acidic residues interacting with the bases and riboses. Two secondary interaction sites involve the long N-terminal domain, which is mainly a coil with few secondary structure elements, and interactions with RNA1 and RNA3 (Fig. 6g). This region is critical for interaction with the SSU, and the stable presence of mtAP2-4 in this position likely plays a crucial role in regulating subunit association. Notably, LSUA is the only rRNA fragment interacting with two mtAP2s (mtAP2-2 and mtAP2-4) simultaneously (Fig. 6f, h). Additionally, the structure of the *T. gondii* mitoribosome is also rich with RNA-binding proteins belonging to RAPs and HPRs families. Four RAPs are components of the LSU, along with five HPRs (four on the LSU and one on the SSU). Unlike the four mtAP2s, which occupy integral positions within the mitoribosome, RAPs and HPRs are predominantly located on the periphery of the solvent side of *T. gondii* mitoribosome (Fig. 6i–k).

### Discussion

AP2 proteins, typically known as transcription factors in plants and Apicomplexa, were previously thought to be exclusively nucleus-

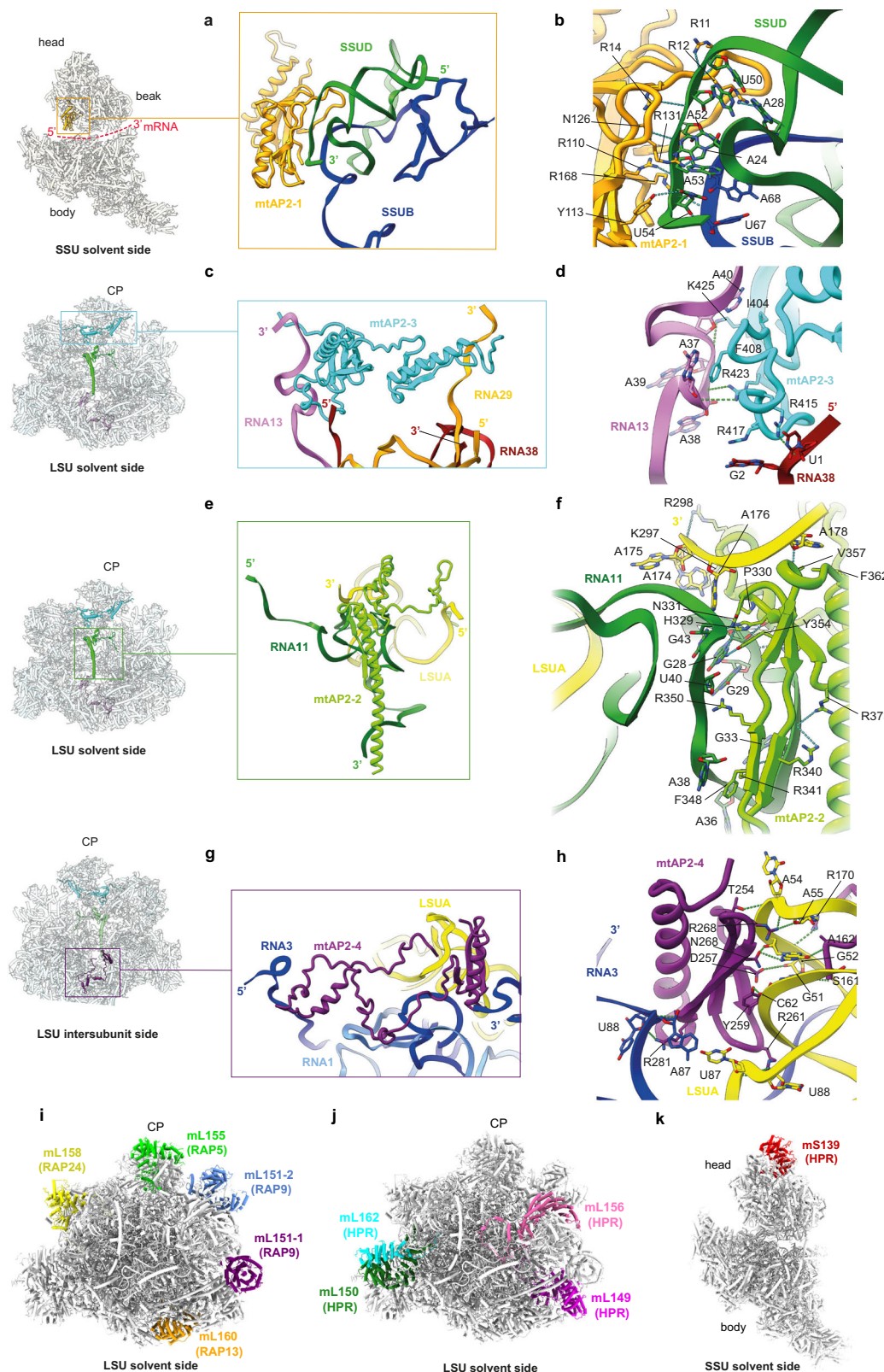

**Fig. 6 | Interaction of the four mtAP2 proteins with the rRNA fragments.**
**a** mtAP2-1 is found in the SSU and localized near the mRNA channel exit (mRNA path is depicted as a broken red line) where it interacts with SSUB and SSUD. **b** mtAP2-1 interacts with the 3' of SSUD and with SSUB mostly through arginine residues. **c** mtAP2-3 protein is located at the central protuberance above the nascent peptide exit channel on the solvent side. mtAP2-3 interacts with the 3' of RNA13, 3' of RNA29 and 5' end of RNA38. **d** Interactions are mediated by arginine residues of mtAP2-3 and the bases of RNA13, RNA29 and RNA38. **e** mtAP2-2 below

the CP, near the NPC, interacts with 3' and 5' of LSUA and is enlaced by RNA11. **f** Interactions are mediated by arginine residues of mtAP2-2 and the bases of RNA11, and 3' end of LSUA. **g** mtAP2-4 protein is located at the LSU intersubunit face and interacts with RNA3 and LSUA. **h** mtAP2-4 interacts primarily with LSUA through arginine and acidic residues that interact with bases of the LSUA. **i**−**k** RAP proteins (**i**) and HPR proteins (**j**, **k**) are found in the peripheral of *Toxoplasma* mitoribosome (gray surface).

localized[23], however, their unexpected presence in the mitochondrion of *T. gondii* is compellingly demonstrated. The four mtAP2s are among the smallest ApiAP2s and yet they massively further truncate into smaller proteins upon maturation (Fig. 1a, b, d). Strikingly, the conservation in all members of the Myzozoa and their absence in Ciliates, which do not display a more conventional mitochondria, provides a strong correlation between the interspecies conservation and mitochondrial rRNAs fragmentation. Unlike previous studies showing that ApiAP2s primarily bind to DNA[39], our data reveal that these mtAP2s are mitochondrial ribosomal proteins that interact tightly with rRNA. The functional significance of these mtAP2 proteins closely parallels that of the mitoribosomal proteins conserved with bacterial homologs identified in *T. gondii* and *P. falciparum*[18–22]. Remarkably, the four mtAP2s were found near functional regions of mitoribosome, indicating roles beyond the structural assembly of rRNA fragments. Additionally, the ability of an *E. maxima* AP2 domain to complement the function of mtAP2-2 in *T. gondii* highlights functional conservation across the Apicomplexa. It is plausible that the four mtAP2s shifted from their potential ancestral role in DNA binding and gene expression regulation to orchestrating the assembly of fragmented rRNAs into functional mitoribosomes. Alternatively, this could indicate that AP2s originally functioned as RNA-binding and evolved into DNA-binding proteins. Collectively, our findings reshape our understanding of the AP2 proteins and other transcription factor families, suggesting a fundamentally different mode of action of transcription factors in general.

The size and number of mt-rRNAs vary significantly among organisms. While Ciliates[37] and green algae[38,40] possess fragmented mt-rRNA genes to some extent, apicomplexans have undergone both genome reduction and extensive fragmentation. Previously, a total of 39 small mitochondrial RNA pieces were reported in *P. falciparum*, with 27 of them assigned to specific regions of SSU and LSU[14]. Recently, 34 small mitochondrial RNA fragments have been identified in *T. gondii*[17]. Our results reveal the structure and composition of the most fragmented rRNA mitoribosome to date.

We observe how evolutionary changes in the mt-rRNA have necessitated the recruitment of numerous species-specific proteins, repurposing members of RNA-binding protein families such as HPRs and RAPs, and reusing of same rRNA fragments in both SSU and LSU to assemble a functional mitoribosme. Besides the five RAPs and five HPRs that are associated with the *T. gondii* mitoribosome, it would be safe to assume that the other mitochondrial RAPs and HPRs are more likely to be implicated in other RNA metabolism functions such as rRNA processing and/or modification of tRNAs.

This provides a basis for future studies on ribosomal complex biogenesis. Despite this divergence, the structure of the first apicomplexan parasite shows that the key catalytic rRNA regions of the mitoribosome, including the decoding center, PTC, and NPC, remain conserved with bacterial ribosomes. Most clinically used antibiotics specifically target these regions[41,42]. Our structure will serve as a foundation for optimizing lead candidates with enhanced selectivity and potency against these human pathogens. From an RNA evolution standpoint, the myzozoan mitoribosome exemplifies extreme species-specificity and underscores the fundamental crosstalk between the nucleus and the mitochondrion.

## Methods

### Animal ethics statement
All animal experiments were conducted with the authorization numbers GE-41-17, according to the guidelines and regulations issued by the Swiss Federal Veterinary Office.

### Mice
Female, 6-week-old BALB/c mice and CD1 outbred mice were obtained from Charles River laboratories. Mice were specific pathogen-free (including *Mycoplasma pulmonis*) and subjected to regular pathogen monitoring by sentinel screening. They were housed in individually ventilated cages furnished with a cardboard mouse house and Nestlet, maintained at $21 \pm 2\,°C$ and 35% humidity under a 12 h light/dark cycle and given commercially prepared autoclaved dry rodent diet and water ad libitum.

### Parasite culture
Tachyzoites of *T. gondii* strains RHΔ*ku80*Δ*hxgprt* and DiCre RH[32] were propagated in confluent HFF monolayers (ATCC SCRC-1041) maintained in Dulbecco's modified Eagle's medium (DMEM, Macgene) supplemented with 2% fetal bovine serum (FBS, Sigma-Aldrich), 10 units/ml penicillin, and 100 mg/ml streptomycin. The infected cells were cultured under 5% $CO_2$ at 37 °C.

For *Plasmodium*, parental cell lines were derived from *P. berghei* ANKA strain-derived clone 2.34. Parental cell lines together with derived transgenic lines were grown and maintained in CD1 outbred mice. The parasitaemia of infected animals was determined by microscopy of methanol-fixed and Giemsa-stained thin blood smears. For gametocyte production, parasites were grown in mice that had been phenyl hydrazine treated three days before infection. Exflagellation was induced in exflagellation medium (RPMI 1640 containing 25 mM HEPES, 4 mM sodium bicarbonate, 5% fetal calf serum (FCS, 100 mM xanthurenic acid, pH 7.8). For gametocyte purification, parasites were harvested in suspended animation medium (SA; RPMI 1640 containing 25 mM HEPES, 5% FCS, 4 mM sodium bicarbonate, pH 7.2) and separated from uninfected erythrocytes on a Histodenz cushion made from 48% of a Histodenz stock (27.6% [w/v] Histodenz [Sigma/ Alere Technologies] in 5.0 mM TrisHCl, 3.0 mM KCl, 0.3 mM EDTA, pH 7.20) and 52% SA, final pH 7.2. Gametocytes were harvested from the interface.

### Antibodies
The following primary antibodies were used in immunofluorescence, immunoblotting, co-immunoprecipitation, RIP, and mitoribosome purification assays: rabbit anti-TgGAP45 (1:10000)[43], rabbit anti-HSP70 (1:1000)[44], mouse anti-Actin (1:100)[45], mouse anti-catalase (1:2,000)[46], mouse anti-Ty (1:200, BB2), mouse anti-HA (1:1000, Sigma-Aldrich, RRID: AB_262051) and mouse anti-FLAG (1:1000, Sigma-Aldrich, RRID: AB_262044). Secondary antibodies for immunofluorescence included Cy3/FITC-conjugated goat anti-mouse IgG(H + L) (1:200, Proteintech SA00009-1/SA00003-1) or Cy3/FITC-conjugated goat anti-rabbit IgG(H + L) (1:200, Proteintech SA00009-2/SA00003-2). For western blotting, secondary antibodies used were HRP-conjugated goat anti-mouse/rabbit IgG (1:1000, Macgene IS001/IS003).

### Bioinformatic analyses
For all proteins involved in this work, the sequences were downloaded from ToxoDB (https://toxodb.org/toxo/app)[47]. Functional domains were predicted in SMART. Protein sequences homologous to the four mtAP2s or mitochondrial ribosomal proteins were obtained from VEuPathDB (https://veupathdb.org/veupathdb/app)[48] or UniProt (https://www.uniprot.org/). The sequences of the four mtAP2s and their homologous proteins were aligned using ClustalW[49]. The phylogenetic tree was generated by neighbor-joining, and the resulting tree in Newick format was visualized using MEGA 11.

### Parasite transfection
For *T. gondii*, around $10^7$ mechanically released parasites were centrifuged at 400 x g for 10 min. The supernatant was removed, and the parasite pellet was resuspended in 500 µL Cytomix buffer. 30 µg of purified plasmid and amplicon DNA were then added to a final volume of 800 µL. The parasites were transferred to a 0.4 cm gap cuvette and electroporated with 1.5 kV at 25 µFd and 50 Ω with the Gene Pulser Xcell electroporation system (BioRad, USA). Transfected parasites were then inoculated onto HFF monolayers and selected with drug

after 24 h post-infection. The hypoxanthine xanthine-guanine phosphoribosyl transferase (HXGPRT)[50], dihydrofolate reductase-thymidylate synthase (DHFR-TS*)[51] and *Escherichia coli* chloramphenicol acetyltransferase (CAT)[52] were used as positively selectable markers, uracil phosphoribosyl transferase (UPRT)[53] as a negatively selectable marker, as described previously.

For *P. berghei*, schizonts for transfection were purified from overnight in vitro culture on a Histodenz cushion made from 55% of the Histodenz stock and 45% PBS. Parasites were harvested from the interface and collected by centrifugation at 500 x g for 3 min, resuspended in 25 μl Amaxa Basic Parasite Nucleofector solution (Lonza) and added to 10 μg DNA dissolved in 10 μl H₂O. Cells were electroporated using the FI-115 program of the Amaxa Nucleofector 4D. Transfected parasites were resuspended in 200 ml fresh RBCs and injected intraperitoneally into mice. Parasite selection with 0.07 mg/mL pyrimethamine (Sigma-Aldrich) in the drinking water (pH ~4.5) was initiated one day after infection.

## Parasite strains generation

The primers and plasmids used or constructed in this study are listed in Supplementary Data 1. The detailed construction methods are as follows:

**mtAP2s tagging in *T. gondii*:** For the CRISPR/Cas9-mediated C-terminal epitope tagging of four mtAP2 proteins, we generated a specific single-guide RNA (sgRNA) to introduce a double-stranded break near the translation stop codon in the 3'UTR. We utilized 59 bp PCR primers with 42 bp homology sequences upstream of the gene's translation stop codon and downstream of the Cas9 break site to amplified PCR products from plasmids containing either 6Ty-HX, 6HA-DHFR-TS* or 3FLAG-CAT. The specific-sgRNA CRISPR/Cas9 plasmid and PCR products were co-transfected into RH Δku80Δhxgprt parasites, followed by selection with appropriate drugs. Clonal strains were isolated from single plaques through limiting dilution. The confirmation was done using IFA and western blotting.

**mtAP2s tagging in *P. berghei*:** The oligonucleotides used to generate transgenic parasite lines are in Supplementary Data 1. C-terminal tagging of *P. berghei* proteins by PCP was as described[54] and targeted endogenous loci by allele replacement. Briefly, sequences comprising ~500 bp from the C-terminus of the coding sequence and ~500 bp from the immediate 3' UTR for genes encoding mtAP2-1/2/3 were cloned into KpnI and XhoI sites upstream to the coding sequences of mNeonGreen fused to triple hemagglutinin epitope tag in pCP-mNG-3xHA, along with a NotI linearization site between the targeting sequences.

**Generation of Split-GFP-related parasites:** For split-GFP assay, we first tagged mtAP2s at their C-terminus with 7×GFP₁₁ using the CRISPR/Cas9 system. Stably transfected parasites were obtained through drug selection with 25 μg/mL mycophenolic acid and 50 μg/mL xanthine, using the HXGPRT drug selectable marker present on the plasmids. We then generated parasite lines expressing the HSP60 MTS sequence (MTS-HSP60) fused with the C-terminal fragment β-strand 1-10 of GFP (MTS-HSP60-GFP₁₋₁₀) in the mtAP2-1 (or 2 or 3)−7×GFP₁₁ background. Stably transfected parasites were selected with 3 μM pyrimethamine, using the DHFR-TS* drug selectable marker present on the plasmids.

**CRISPR-Cas9 mediated gene knock-down (TATi-TetO7 and 4U1/DiCre):** To introduce an ATc-regulated promoter into the mtAP2s and uL4m, CRISPR/Cas9 plasmids target the region near the start codon for each specific gene were generated. The plasmids were then co-transfected with purified PCR products containing the TATi-1 transactivator, the HXGPRT selectable cassette, and a TetO7SAG1 promoter into the subsequent parasite (mtAP2-(1-3)-Ty, uL4m-FLAG). For the generation of mtAP2-1-Ty-U1 and mtAP2-3-Ty-U1 inducible knockdown strains. DiCre RH parasites were transfected with CRISPR/Cas9 plasmid designed for mtAP2s tagging and purified PCR products containing a 6Ty-4U1-HXGPRT cassette. Stably transfected parasites

were selected using the HXGPRT drug selectable marker. Clonal strains were isolated from single plaques by limiting dilution and verified through IFA and western blotting.

**mtAP2-2 deletion in *T. gondii*:** To delete the mtAP2-2, a CRISPR/Cas9 plasmid with sgRNA targeting the middle of its coding region was co-transfected with a gene-specific knockout plasmid containing a DHFR-TS* selectable marker flanked by two gene-specific 5' and 3' homology arms (around 800 bp each). Transfected parasites were selected with pyrimethamine 24 h post electroporation and recovery. Clonal strains were isolated from single plaques by limiting dilution and subsequently confirmed by diagnostic PCR. See Supplementary Fig. 2i for a detailed illustration of the gene knockout strategy.

**mtAP2s complemented parasites:** C-terminally 6×HA-tagged cDNAs corresponding to full length or truncated versions of the four mtAP2s were generated. The upstream and downstream homology regions required for integration at the UPRT site were amplified from genomic DNA of *T. gondii*. The four mtAP2s-regulatable parasites were then co-transfected with CRISPR/Cas9 (UPRT, CDS) plasmids and their corresponding complementation plasmids. Parasites were selected in 5 μM 5-fluorodeoxyuridine (FUDR) containing medium and the clonal strains were confirmed through IFA and western blotting.

**mETC subunit and mitoribosomal subunit proteins tagging:** To generate tagging lines for mETC subunit proteins (SDHB, OCR12, Cox26 and F1γ) and mitoribosomal subunit proteins (uL4m, bL27m, bL21m, uL3m, uS5m and bS21m) in three mtAP2s HA-tagged lines (mtAP2-1-HA, mtAP2-2-HA and mtAP2-3-HA) or four mtAP2s knockdown lines, we introduced a 3×FLAG or 6×Ty at the C-terminus of these proteins using a CRISPR/Cas9 plasmid with a specific sgRNA targeting the region near the stop codon of the respective proteins. The CRISPR/Cas9 plasmid was co-transfected with purified PCR products containing the 3×FLAG tag and the CAT selectable cassette into the corresponding parasites. Stably transfected parasites were selected with chloramphenicol (34 mg/ml).

## Western blotting

Protein samples were lysed in RIPA buffer (50 mM Tris pH 7.4, 150 mM NaCl, 1% Triton X-100, 1% sodium deoxycholate, 0.1% SDS; Beyotime) supplemented with 1 mM PMSF protease inhibitor. The samples were then separated by SDS-PAGE and transferred onto nitrocellulose membranes. Incubation with primary and secondary antibodies was performed for 1 h each at room temperature, and protein visualization was achieved using a chemiluminescence reaction. Quantification of protein abundance was performed in ImageJ.

## Protein localization for *T. gondii*

Freshly harvested parasites were inoculated on HFF monolayers grown on glass coverslips in 12-well plates. The infected cells were fixed with 4% paraformaldehyde (PFA) for 30 min, permeabilized with 0.25% Triton X-100 in PBS for 30 min, and then blocked with 3% bovine serum albumin (BSA) for an additional 30 min. Subsequently, the coverslips were incubated with primary antibodies for 1 h, followed by three PBS washes. Nuclei were stained with Hoechst 33258 (1:100 dilution) and secondary antibodies for 1 h, followed by three PBS washes. Experiments were conducted at 37 °C or room temperature. Infected monolayers were examined using a Leica confocal microscope (Leica, TCS SP52, Germany) with 63x magnification. High-content imaging and analysis were performed with the LAS AF lite 2.2.0 software.

## Protein localization for *P. berghei*

For localization of tagged proteins in *P. berghei* by native fluorescence, cells from different proliferative stages during the parasite lifecycle were mounted in exflagellation medium (RPMI 1640 containing 25 mM HEPES, 4 mM sodium bicarbonate, 5% fetal calf serum (FCS), 100 mM xanthurenic acid, pH 7.8), in the presence or absence of MitoTracker™

Red CMXRos and Hoechst 33342 and imaged using an inverted Zeiss Axio Observer Z1 microscope fitted with an Axiocam 506 mono 14 bit camera and Plan Apochromat 63x / 1.4 Oil DIC III objective. All images of fluorescent proteins were captured at RT with equal exposure settings. Images for level comparison were processed with the same alterations to minimum and maximum display levels. Analysis and displays were performed in Fiji and the statistical programming package R.

### Plaque assay

Freshly egressed parasites, at a density of 70 parasites per well in 12-well plates, were inoculated onto HFF monolayers. Parasites were grown for 7 days before fixation with 4% paraformaldehyde in PBS and stained with 0.1% crystal violet. Plaque formation was quantified by measuring the areas of clearance created by the parasites. For Tet-repressive knockdown strains, anhydrotetracycline (ATc, 1 μg/ml) or vehicle (ethanol, 1:2000) was additionally added during inoculation. To evaluate the reversibility of four mtAP2s knockdowns, parasites were either preincubated in ATc for 3 days or not before conducting the plaque assay. In the case of DiCre-mediated knockdown strains, infected cells were treated with 50 nM rapamycin or DMSO for 4 h before washing out.

### Replication assay

Approximately $10^5$ freshly harvested parasites were inoculated onto HFF monolayers grown on glass coverslips in 12-well plates for 1 h in standard media. Subsequently, the monolayers were washed three times with PBS to remove the non-invaded parasites. After 24 or 48 h post-infection, the infected monolayers were fixed in 4% paraformaldehyde (PFA) and then stained with anti-GAP45 (1/5000) antibodies. Parasite quantification per vacuole was conducted by analyzing 100 vacuoles for each experimental condition. Three independent experiments were performed.

### Virulence assay

Six-week-old female BALB/c mice were inoculated with $10^4$ ΔmtAP2-2, ΔmtAP2-2::mtAP2-2, or RH Δku80 tachyzoites (5 mice per group). The survival of the mice was monitored for one month. Mice that survived were subsequently challenged with $10^5$ with RH Δku80 tachyzoites and monitored for an additional month. Non-immunized naïve mice, receiving the same challenging doses, served as the control group.

### Transmission electron microscopy (TEM)

Parasites were fixed overnight at 4 °C in a freshly prepared solution of 2.5% glutaraldehyde in 100 mM phosphate buffer (pH 7.4). Subsequently, they underwent two rinses, first with PBS and then with dH₂O. The samples were then immersed in an aqueous solution containing 1% osmium tetroxide and 1.5% potassium ferricyanide at 4 °C for 2 h. Following several rinses in dH₂O, dehydration was performed using a graded series of ethanol (30%, 50%, 70%, 80%, 90%, 100%, 100%, 10 min each) and pure acetone (10 min, two times). Next, the samples were infiltrated with a graded mixture of acetone and SPI-PON812 resin (3:1, 1:1, 1:3), followed by pure resin. Finally, the samples were embedded in pure resin with 1.5% N, N-dimethylbenzylamine (BDMA) and polymerized for 12 h at 45 °C or 48 h at 60 °C. The ultra-thin sections of 70 nm were cut with a microtome (Leica EM UC6), double-stained with uranyl acetate and lead citrate, and viewed on a transmission electron microscope (FEI Tecnai Spirit 120 kV).

### Extracellular flux analysis with Seahorse

Freshly egressing parasites of the tested strains treated with ATc (1 μg/ml) for 72 h or not were harvested and filter-purified (3 μm exclusion size). The assay was performed as previously described[33,55] with few modifications as outlined here. The pellet of purified parasites was washed with Seahorse XF medium (Agilent, USA) supplemented with 10 mM glucose, 2 mM glutamine, 1 mM sodium pyruvate and ATc (1 μg/mL), adjusted to pH 7.4. Subsequently, $10^6$ parasites were seeded in 180 μl of XF medium as above per well in a 96-well plate, coated with CellTak (Corning, USA) according to the manufacturer's guidelines. All samples were measured using an Extracellular Flux Analyzer XF-96 (Seahorse, Agilent, USA). Mixing and measuring were performed in three cycles for the basal respiration and after addition of oligomycin (final concentration of 2.5 μM), carbonyl cyanide-p-trifluoromethoxyphenylhydrazone (FCCP, final concentration of 2 μM) and antimycin A (0.5 μM). For Fig. 2d, the oxygen consumption rate and extracellular acidification rate of the first three measurements (basal) were averaged. The data show the mean (dots) ± SD of three independent biological experiments, each consisting of >18 technical replicates. A one-way ANOVA, followed by Bonferroni multiple pairwise comparison was performed testing for statistical differences between +/− ATc treatments. For Fig. 2e, datapoints and error bars show the mean ± SD of a single representative experiment performed three times, consisting of >12 technical replicates. Calculations and statistical analyses were performed using Wave (Agilent), Excel (Microsoft) and Prism (GraphPad) Software.

### RNA extraction and RNA-Seq

Freshly egressed parasites were simultaneously inoculated on HFF monolayers growing in T-75 flasks and were washed to remove non-invaded parasites after 1 h post-infection. Infected cells were then treated with ATc or vehicle for 48 or 60 h and subsequently released from host cells. Total RNAs were extracted using Trizol® and genomic DNA removal using DNase I (Qiagen, Hilden, Germany). Sequencing was conducted using an Illumina HiSeq-PE150, and raw reads were filtered with Trimmomatic-0.38. Clean reads were mapped to the *T. gondii* reference genome (ToxoDB62_TgondiiGT1_Genome.fasta and ToxoDB-62_TgondiiGT1.gff)[47] using the HISAT2[56]. Each condition was replicated three times biologically. Differential expression analysis was performed using the DESeq2 R package, and adjusted *P*-values were calculated using the Benjamini & Hochberg method. A log2 (fold change) of 1 was set as the threshold for differentially expressed genes and *P*-values were adjusted to 0.01.

### Quantitative label-free mass spectrometry

Each protein sample was obtained from approximately $10^8$ *Toxoplasma* tachyzoites and quantified using the Bradford assay. 50 μg of proteins were separated and DTT was added (final concentration of 10 mM) and incubated at 37 °C for 1 h. After additional incubation with iodoacetamide for 30 min at room temperature, all samples were diluted 4-fold with 25 mM ammonium bicarbonate (ABC) buffer and digested with trypsin (at 1:50 trypsin: protein concentration) at 37 °C overnight. The digested peptides were recovered, desalted, and concentrated for detection. Protein samples were submitted to Beijing Qinglian Biotech Co., Ltd. for mass spectrometry. The eluted peptides were analyzed by Q Exactive HF-X mass spectrometer (Thermo Fisher Scientific). The raw files were analyzed against the ToxoDB *Toxoplasma* Genomics Resource (ToxoDB-56_TgondiiGT1_AnnotatedProteins.fasta)[47] using MaxQuant software. The current *T. gondii* genome assembly does not contain the three mitochondrial protein-coding genes (cob, cox1 and cox3) and mtAP2-4, so we assembled these four protein sequences into the annotated protein file for subsequent analysis. The differentially expressed proteins with >1.5-fold change and B−H adjusted *P*-values < 0.05 were analyzed. Statistical significance was calculated by unpaired two-tailed Student's *t*-test.

### Immunoprecipitation and Co-immunoprecipitation

To isolate dissociated mitoribosomes, freshly lysed parasites from four T-75 flasks of monkey VERO cells were collected and lysed in cold IP buffer (50 mM Tris, pH 7.4, 150 mM NaCl, 1% NP-40, protease inhibitor

cocktail). For mature mitoribosomes, freshly lysed parasites were pelleted by centrifugation at 4 °C with 100 µg/mL cycloheximide (CHX) and 100 µg/mL Chloramphenicol (CAP). Purified parasites were lysed in mitoribosome lysis buffer (50 mM Tris-HCl, pH 7.5, 150 mM NaCl, 10 mM MgCl₂, 1% NP-40, 0.5% Triton X-100), freshly supplemented with 1 mM DTT, 100 µg/mL CHX, 100 µg/mL CAP, 2 mg/ml complete protease inhibitor cocktail, and 0.04 unit/ml of RNAsin. Following four cycles of freeze/thaw, the lysate was sonicated on ice for 3 min total pulse time (2 s ON, 3 s OFF and 20% power). Sonicated samples were then centrifugated 15 min at 16,000 × g at 4 °C and incubated with mouse anti-HA antibodies (Sigma-Aldrich), anti-FLAG antibodies (Sigma-Aldrich) or anti-Ty antibodies overnight at 4 °C. Following overnight incubation, protein A/G magnetic beads (MCE) were added and incubated for an additional 3 h. Beads were then washed four times in IP buffer or mitoribosome buffer and two times in PBS. For mass spectrometry analysis, 10% beads were resuspended in SDS-PAGE sample buffer and detected by Western blotting. The remaining fraction of the beads were then loaded onto the SDS-PAGE gel and bands were cut out for mass spectrometry analysis. For the co-immunoprecipitation experiments, freshly lysed parasites were purified, and lysed in cold IP lysis buffer or mitoribosome lysis buffer. The following experimental procedure was consistent with the IP described above. The sonicated samples were both incubated with anti-FLAG antibodies (anti-Ty antibodies) and anti-HA antibodies, respectively and analyzed by Western blotting.

## LC-MS/MS acquisition and data analysis

Samples were digested with trypsin and the resulting peptides were desalted on C18 Cartridges (Empore™ SPE Cartridges C18 (standard density), bed I.D. 7 mm, volume 3 ml, Sigma) and reconstituted in 40 µL of 0.1% (v/v) formic acid. LC-MS/MS analysis was performed on the Nanoelute (Bruker) coupled to timsTOF Pro mass spectrometry (Bruker). Precursors and fragments were analyzed with the TOF detector, and full mass scans were acquired over a range of m/z 100-1700. The MS raw files were processed with MaxQuant (version 1.6.14) for peptide identification and quantitation analysis. The files were searched against the *T. gondii* database (ToxoDB-66_TgondiiME49_AnnotatedProteins)[47]. Three mitochondrial-encoded genes and mtAP2-4 were included. Quantitative analysis was performed as described previously[57]. *P*-values and log2 (fold change) significance cutoffs for IP experiments were set appropriately.

## RIP-Seq

Four T-75 flasks of freshly egressed RH *ku80Δhxgprt* or mtAP2-(1-3)-HA parasites were harvested. Parasite pellets were then lysed in cold IP buffer or mitoribosome buffer. After undergoing four cycles of freeze/thaw, the lysates were additionally sonicated on ice for a total pulse time of 3 min (2 seconds ON, 3 seconds OFF and 20% power). The sonicated samples were then centrifugated 15 min at 16,000 × g at 4 °C. 10% of the lysate was reserved as "Input samples". The remaining volume was subjected to IP with 5 µg mouse anti-HA antibodies. Following overnight incubation, protein A/G magnetic beads (MCE) were added and incubated for an additional 3 h. Beads were then washed four times with IP buffer or mitoribosome buffer and two times with PBS. 10% of the beads were used for Western blotting analysis. The remaining beads and input samples were then RNA extracted using Trizol®. RNA purified from both input and IP samples was concentrated by ethanol precipitation and resuspended in equivalent volumes of RNase-free water. Libraries from both input and IP samples were prepared using KC-Digital™ Stranded mRNA Library Prep Kit (RNA-Seq) or QIAseq® miRNA UDI Library Kit (QIAGEN, small RNA-Seq) following the manufacturer's instruction. Input and IP samples are not subjected to poly(A) selection to preserve the integrity of mt-rRNA. Sequencing was conducted using the Novaseq 6000 (Illumina Inc) in PE150 mode.

## Analysis of mitoribosomal RNA pieces

For the standard RNA-Seq, pulldown (IP) and corresponding input reads were mapped by HISAT2[56] with default parameters using the *T. gondii* genome and gtf transcripts definitions taken from release 48 of the ToxoDB[47] annotation, along with 23 custom code blocks (A-V) and prevalent sequence block combinations (MT-1/2/3) for the mitochondrial genome[17], as illustrated in the relevant figures (Supplementary Fig. 5b, c). Within the custom code blocks, depth values were calculated for the middle point of reads at each position and then averaged every 10 bases across the block. The resulting values were normalized to the total counts, genome-wide for their respective samples giving counts per million (CPM) for each bin overall IP and input replicates. To produce binding enrichment levels for each pulldown, a log2 fold change is taken for each bin of (1 + IP mean)/(1 + INPUT mean). Up to this point, enrichment values for the small RNA-Seq were calculated identically, save for the addition of extra sequence block combinations into the mitochondrial genome for mapping, as indicated in the figures (Supplementary Fig. 5d). Finally, a further filter was imposed for the inclusion in the enrichment plots of the standard RNA-Seq - the enrichment had to be greater than 2 on either the positive or negative strand for two consecutive bins to be included in the plot, otherwise it was set to 0. This filter was not included for the small RNA-Seq to permit smaller binding regions. Alignments and coverage graphs were generated and visualized using the Integrative Genomics Viewer with the acquired bedgraph files (Supplementary Data 6).

## Mitoribosome fractionation

Extracellular parasites from a 150 mm dish grown on HFF cells, treated with ATc (+ ATc) or without (- ATc), were pelleted by centrifugation at 4 °C in the presence of CHX and CAP. Cells were lysed in mitoribosome lysis buffer (described above) by 4 rounds of freezing and thawing, followed by sonication (5 ×20 pulses at 50% duty cycle, output control 2) using a probe sonicator. Lysates were precleared by centrifugation at 16,000 × g at 4 °C. Subsequently, 2 mg of total extracts (TE) were loaded onto a 10-50% sucrose gradient prepared in mitoribosome lysis buffer but without detergents. The gradient was spun in a SW 41 rotor (Beckman) at 288,000 x *g* for 2.5 h at 4 °C. Fractions were collected using the gradient fractionator (Teledyne Isco) and then precipitated with trichloroacetic acid (TCA). The distribution of mitoribosomal subunits in 10–50% sucrose gradients was examined under wild-type conditions (+ mtAP2) compared to mtAP2 depletion (- mtAP2). Fractions were collected from the top and subjected to western blot analysis. Fraction 1 represented the lightest fraction, while fraction 14 was the heaviest.

## RNA isolation and reverse transcription

Fractions 1 to 14 from the sucrose gradients were harvested, and RNA was isolated by adding 0.7 volumes of TRIzol (Invitrogen) and 0.3 volumes of chloroform. The mixture was vortexed and centrifuged at 20,000 × g for 15 min at 4 °C. The upper phase was collected, and the RNA was precipitated with isopropanol and Na-Acetate. The resulting pellets were resuspended in Milli-Q water. After treatment with DNase I (QIAGEN), a fraction of the RNA (50–100 ng) was subjected to reverse transcription (RT) using M-MLV reverse transcriptase (Promega) and either random primers (mt-rRNA) or oligo dT primer (TgACT). The RT reaction included 80 units/ml RNase inhibitor RNasin (BioLabs) according to the manufacturer's instructions. Real-time PCR quantification was performed using 1/50 of the cDNA product and specific primer pairs for the indicated genes. The obtained values were normalized to those obtained for the control mRNA, as indicated in Fig. 3e, to account for differences in total RNA among samples.

## Mitoribosome immunopurification

250 plates (150 mm) of freshly extracellular parasites expressing FLAG-tagged uL4m were grown onto monkey VERO cells monolayers were

harvested. The pellet was lysed in mitoribosome lysis buffer (described above). The supernatant was cleared and then incubated with ANTI-FLAG M2 Affinity Gel (Sigma-Aldrich) overnight at 4 °C. Subsequently, it was washed three times with mitoribosome lysis buffer (10 times beads volume) and eluted with 3×FLAG peptide (MCE) prepared in the same buffer (final concentration of 300 ng/ml). The elution was loaded onto 10-50% sucrose gradients, and the fractions corresponding to LSU, and monosome were isolated and concentrated by ultra-centrifugation at 300,000 × g for 3 h at 4 °C. The resulting pellets were washed and resuspended in monosome buffer (5 mM HEPES-KOH pH 7.5, 100 mM KCl, 20 mM MgOAc, 5 mM NH$_4$Cl, 0.5 mM CaCl$_2$, 1 mM DTT, 1 mM spermidine).

### Sample preparation for Cryo-EM

Sucrose fractions containing mitochondrial full mitoribosome or LSU or monosome were centrifuged at 800,000 × g for 3 h at 4 °C (S140-AT fixed angel rotor) and the pellets were dissolved in 10 to 20 µL of buffer containing 20 mM HEPES-KOH pH 7.6, 100 mM KCl, 20 mM MgCl$_2$, 1 mM DTT, 0.02% DDM and Roche complete protease inhibitor cocktail.

### Cryo-EM grid preparation

4 µL of the LSU or monosome sample at a molar concentration of 165 and 50 nM, respectively, were used to apply onto Quantifoil R2/2 300 mesh holey carbon grid coated with thin home-made carbon film of 2 nm and glow-discharged at 2.5 mA for 25 s. The sample was incubated on the grid for 30 s and then blotted with filter paper from both sides for 1.5 s with the blot force 5 at 4 °C and 100% humidity controlled by Vitrobot Mark IV, followed by grid vitrification in liquid ethane pre-cooled by liquid nitrogen.

### Data acquisition

The data acquisition was performed on a Talos Arctica instrument (FEI) operated at 200 kV acceleration voltage and at a nominal underfocus of Δz = -0.5 to -3.0 µm using the CMOS Summit K2 direct electron detector 4096 × 4096 camera and automated data collection with SerialEM at a calibrated magnification of 45,000 x. The pixel size of 1.16 Å at the specimen level (the calibrated magnification on the 6.35 µm pixel camera is 115,455 x). The camera was set up to collect 40 frames that were subsequently realigned. Total collected dose is -50 e-/Å2.

### Image processing

MotionCor[58] was used for the movie alignment of the movie frames from all datasets. CTFFIND4[59] was used for the estimation of the contrast transfer function of an average image of the whole stack. Particles selection and further particle processing were performed in CryoSPARC[60,61]. 9524 movie-stacks were acquired for the LSU dataset (Supplementary Fig. 7a, b). 16741 movie-stacks were acquired for the full mitoribosome sample (Supplementary Fig. 7e, f). After particle sorting through 2D classification, selected classes were used to run an ab initio reconstruction in CryoSPARC, followed by 3D classification. Selected classes were pooled together and used to run a Non-uniform refinement[61]. At the end of the processing pipeline (Supplementary Fig. 7b), the highly homogenous ensemble of particles belonging to exclusively one class presenting the LSU yielded reconstruction at a resolution of 2.89 Å from 206505 particles (Supplementary Fig. 7c, d). Similar processing pipeline was applied for the full mitoribosome sample, which after 2D and 3D classifications yielded two structures, a dissociated LSU and the full mitoribosome. The mitoribosome was derived from only 22169 particles (Supplementary Fig. 7f) by Non-uniform refinement and then further locally refined to three bodies, the SSU head, the SSU body and the LSU, at resolutions of 3.6 Å, 3.29 Å and 3.11 Å, respectively (Supplementary Fig. 7g–j). The dissociated LSU was derived from 189305, thus representing more than 88% of the final particles

count. The final cryo-EM reconstructions were also enhanced using DeepEMancer[62] and were used only as display items for Fig. 5.

### Atomic model building, 2D rRNA mapping and refinement

For modeling the *T. gondii* mitochondrial ribosome structure, three high-resolution maps including LSU, SSU head and SSU body were used to assign the atomic coordinates. The maps were subjected to automatic modeling using ModelAngelo v1.0.10 with no-sequence mode[63]. The assignment and connection of the protein fragments were accomplished by aligning each Hidden Markov Models (HMM) fragment longer than 10 amino acids with the Mass-Spectrometry data of *T. gondii* mitoribosome purification by using the Python library PyHMMER v0.10.12[64]. In the case of unfound hits, the alignments of these protein fragments with the *T. gondii* proteome (Proteome ID UP000001529) were performed. The AlphaFold2 models of the assigned proteins were then structurally aligned to their counterparts in the model to map the unsolved protein fragments[63,65,66]. Subsequently, the amino acid sequences in the atomic model were corrected by serial point mutations using the Biobb_model v4.1 Python library (https://github.com/bioexcel/biobb_structure_checking).

The *T. gondii* mitochondrial rRNA fragments were assigned in the atomic models through a combination of visual inspection, nucleotide sequence alignment, and mapping their secondary structure on the *E. coli* 50S model (PDB ID: 6PJ6), 30S model (PDB ID: 7OE1) and *Plasmodium* rRNA secondary structure maps[14].

The secondary structure diagrams of *T. gondii* mitochondrial rRNAs were mapped on the *E. coli* ribosomal 16S and 23S rRNA secondary structure maps that were obtained from RiboVision suite (http://apollo.chemistry.gatech.edu/RibosomeGallery/)[67]. The 3D mt-rRNA models of *Toxoplasma gondii* were structurally aligned to the 3D rRNA models of *E. coli* 50S and 30S by ChimeraX software[68] and then the aligned rRNA fragments in the *T. gondii* model, and the unalignable rRNA fragments were predicted via RNAfold webserver[69].

The assigned rRNA sequences in the model were then corrected using ChimeraX v1.7[70]. The atomic models underwent stereochemical, torsional, and angle refinement and optimization using Phenix v1.21[71] and ISOLDE v1.6[72]. The refined models were subjected to gradient-driven minimization through five macrocycles and a maximum of 500 iterations using Phenix v1.21[71].

### Quantification and statistical analysis

All data for quantification analyses are presented as mean ± SD. Statistical significance was calculated by unpaired two-tailed Student's t-test or one-way ANOVA, and all analyses were performed with the Graphpad Prism 8.0 software (https://www.graphpad.com/). Comparisons were considered statistically significant when $P < 0.05$. The number of biologically independent replicates (n) and statistical details can be found in the figure legends of both main and supplementary items.

### Reporting summary

Further information on research design is available in the Nature Portfolio Reporting Summary linked to this article.

## Data availability

The electron densities are deposited in the EMDB under the accession codes: EMD-50448, EMD-50470, EMD-50493, and EMD-51104 for the SSU head, SSU body, monosome-derived LSU and the LSU derived from the LSU-only sample, respectively. The molecular models derived from these maps are available under the PDB codes 9FI8, 9FIA, and 9G6K, for the SSU head, SSU body and LSU, respectively. Raw RNA-Seq/RIP-Seq data are publicly available through the following BioProject accession number: PRJNA1033534. Mass spectrometry data have

been deposited to the ProteomeXchange Consortium via the iProX partner repository with the dataset identifier PXD046685 and PXD047073. Source data are provided with this paper.

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

## Acknowledgements

This work was supported by the National Key Research and Development Program of China (grant number 2022YFD1800200 to XS); the National Natural Science Foundation of China (NSFC) (grant number 81971953 to YJ) and Beijing Municipal Natural Science foundation (grant number 7182023 to YJ); the Swiss National Science Foundation (SNSF) (grant number TMAG-3_216166 and SNSF 310030_215445, respectively to DSF); the National Natural Science Foundation of China (NSFC) (grant number 32202828 to CW); the ANR Project ANR-21-CE12-0012 (ARAMIS to YH) and the European Research Council Consolidator Grant SPIC-TRANS (ID: 101088541 to YH). We are grateful to Zhongshuang Lv, Xixia. Li, and Li Wang for helping with electron microscopy sample preparation and taking TEM images at the Center for Biological Imaging (CBI), Institute of Biophysics, Chinese Academy of Science. We are thankful to George Edward Allen for analyzing the RIP-Seq data. We thank Dr. Ernest Abboud for his contributions to the project. We sincerely thank Dr. Moritz Treeck for providing the DiCre RH Δku80 line. We are grateful to Dr. Shaojun Long for providing the plasmids. We are thankful to Dr. Hangjun Ke, Dr. Jinke Cheng, Dr. Qun He, Dr. Weiquan Liu, Dr. Jun Tu and Dr. Qian Zhao for their helpful discussion.

## Author contributions

Y.J., D.S.-F., Y.H., X.S. and C.W. conceived the project. Y.J., D.S.-F., Y.H. and X.S. supervised the project. Y.J., D.S.-F., Y.H., X.S., C.W. and S.K. designed the experiments. C.W., S.K., R.E.O.R., P.S., Y.J., J.K., L.B., A.B., O.V., X.L., H.P., X.T., F.X. and Z.W. performed the experiments and analyzed data. Y.H. and F.B. performed cryo-EM experiments. Y.H, R.E.O.R., T.T.N., S.B. and M.L.C. analyzed the cryo-EM data and mapped the rRNA fragments. Y.J., D.S.-F., S.K., Y.H. and C.W. wrote the manuscript. Y.J., D.S.-F., Y.H., S.K., C.W., and X.S. discussed the results and Y.J., D.S.-F., Y.H., S.K., C.W. and X.S. revised the manuscript. All authors read and approved the final manuscript.

## Competing interests

The authors declare no competing interests.
