## [Transparent Peer Review file · Nature Communications]

Apicomplexan Mitoribosome from Highly Fragmented rRNAs to a Functional Machine

Corresponding Author: Dr Yonggen Jia

Version 0:

Reviewer comments:

Reviewer #1

(Remarks to the Author)

This robust and extensive body of work describes the components and structure of the *Toxoplasma gondii* mitoribosome. The structures of this molecular machine are quite different, in comparison to what we know from model organisms, being composed of highly fragmented ribosomal RNAs. There is both high significance and innovation in the manuscript and it should be of impact beyond the field of parasitology. The work presented is of high quality and well executed. Nonetheless, the manuscript is very dense and difficult to follow given the large number of figure panels with little to no description. Streamlining what is presented in the main figures with a focus on the main storyline would aid in making the manuscript more accessible to the readers.

Major

This manuscript is extremely dense in data. The authors should do everything possible to ease its interpretation and reading by simplifying and prioritizing what they include. Following are a few items that could address this issue:

- Including data on *P. berghei* has limited impact, consider not including it or making Plasmodium data a stand-alone supplementary figure.
- If Plasmodium data is kept, be clear in the text and figure legend about whether data is from *Toxoplasma* or *Plasmodium*.
- The experiment using the MPP knockdown (fig 1g and 1h) is clever but superfluous to the main story. Either take it out or make it a supplementary data. Same with data in 1i. This will unclutter Figure 1.
- The experiment using mtAP2-2 and 2-3 tagged with an AID (Auxin Inducible Degron) in Extended Fig1e is not necessary, and it is not mentioned in the text.
- The experiment in extended data 2f with the *Eimeria* homolog is odd as that homolog had not been characterized in terms of function and localization within *Eimeria*. Seems more appropriate to use the *Plasmodium* version to keep a cohesive story in the paper.
- Fig 3 follows the same density of data and looking at the text and the figure is hard to follow through it. The extended data in Fig 3 should also be better explained.

There is a concerning omission of prior published work on the mitoribosome. In particular, the authors should incorporate prior work on their discussion and cite Shikha et al 2022 and Lacombe et al 2019.

Minor

In the main figure 2 (a and b), the amount of data makes it difficult to interpret all the graphs. The colors used by the authors are difficult to distinguish on the screen. I strongly suggest that the authors modify the color scheme for both the proteomics and RNAseq data (Complex II and V, for example).

Figure 3 has the same problem, as the grey-filled dots are difficult to distinguish in the graphs.

Line 102, LOPIT is empirical data and not a prediction.

Line 120 indicate that the fitness data is for *Toxoplasma*

Line 122, for clarity perhaps indicates which ones were knockdown with the tet inducible system and which with the snRNP mediated system

In figure 4, the relative position of panel f is not ideal.

In the figure legend for figure 4, (e) appears to be missing after “ and the small ribosomal subunit side”.

Reviewer #2

(Remarks to the Author)

Title: Apicomplexan Mitoribosome from Highly Fragmented rRNAs to a Functional Machine

Summary

Wang et al., report that four ApiAP2s (1-4) localize to the mitochondrion in human parasite *T. gondii*. They further establish that these transcription factor-like proteins are required for the structural integrity of the mitoribosome but not for transcription. With a combination of single particle cryo-EM, RNA sequencing and mass spectrometry, the authors determine the structure of the mitoribosome. This provides the structural basis not only for the role of AP2s in the mitoribosome structure but also how a highly fragmented rRNA (composed of over 56 fragments) forms the mitoribosome. Additionally, the authors also identify RNA-binding RAP and HPR proteins in the structure leading to insights as to how they stabilize the mitoribosome.

This study provides insights into the various structural mechanisms enabling the assembly of mitoribosome from numerous short rRNA fragments. This is an important contribution to our growing understanding of the drivers of mitoribosome evolution. Since, this comes from a human parasite, this is also very useful for future rational drug development efforts. The quality of the data is good and the manuscript reads well. The illustrations are clear and unambiguous. I am happy to have reviewed this manuscript and recommend the publication of this work. However, I would like to make the following recommendations towards the cryo-EM data interpretation and discussion.

As I understand, the number of fragments are determined to be 56 (33 in LSU and 23 in SSU) by a combination of cryo-EM and RNA-seq. However, these include 21 chains which are modeled as ulr/usr as the sequences could not be identified from RNA-seq data. Since, the identity of these have not been established, it cannot be ruled out that some of these are (unmodeled) extensions of the identified rRNA chains. Therefore, I think the number 56 could be an overestimation and the authors could consider rephrasing ‘over 56 fragments’ in the abstract.

The validation report shows sequence discrepancy in the chain HJ which should be addressed.

Lines: 283, 284: “We chose to represent unassigned fragments as poly-U fragments in our deposited structures and to give them generic names based on their provenance “usr” and “ulr” for unassigned SSU and LSU rRNAs, respectively.” However, in the PDB validation reports, I could only find reference to ‘ulr’ chains. Kindly check.

The values of model vs data cross correlation for LSU reported in extended table 1 seem way too low, probably a typo?

It would be great to see a brief comparison between rRNA fragmentation patterns determined from the current work and those published recently, from *Polytomella magna* (10.1038/s41467-022-33582-5) and *Chlamydomonas reinhardtii* (10.1038/s41467-021-27200-z). Panels could be added to Extended data fig. 9.

The following suggestions are directed at the mitoribosome model:

In SSU head, there is a large unmodeled density likely an rRNA helix continuous with chain hG (resi 40-41;). The following resi 44-56 of hG (terminal residue in the chain) form a helix with chain bN (SSUB) resi 74-82, but, do not show the usual Watson-Crick base-pairing with the corresponding residues. This indicates that chain hG is likely not correctly modeled and can be improved so as to accommodate more residues (likely as a helix) in the unmodeled density, using WC-pairing within chain hG and with resi 74-82 of chain bN as a guide.

A large unassigned volume in SSU head, that runs along chain HK (resi 41-65) and continues to fill up the inter-facial cavity between chains HF, HK, HO and HP. This is likely an RNA chain. The authors could perhaps assign this as a Poly-U chain if sequence can not be determined.

In SSU head, two large unmodeled densities (very likely protein) closely bound to chain HR.

A) One looks like a helix-loop-helix likely to be the missing residues from chain B4 (resi 43-94).

B) The other could be a part of chain Yq. However, it seems to be weakly connected (at lower threshold) with another unmodeled density sandwiched between chains Br and Yi.

Authors could consider modeling these as poly-Ala if the sequence identity could not be determined as the backbone can still probably be traced.

At SSU-LSU interface, there is an unmodeled density that connects RNA chain bS (terminal) to RNA chain bE (at residue 94-95). Following resi 95 of chain bE, the sequence does not appear to match the density very well. For example, 1) resi 96-99 are completely outside the density; 2) resi 103 and 104 are modeled as U and C, respectively, however, the density suggests that these should be purines.

The density currently assigned to resi 95-104 of chain bE, in my interpretation, is likely a part of the chain bS.

In the LSU, at the interface between chains L4 and Lh there is a density potentially corresponding to 6-7 nucleotides. At this resolution it looks like a poly-purine stretch. Additionally, chains, lU (LSU) and bY (SSU) built as poly-uridine look (from the density) almost entirely like long poly-purine chains. They could be considered to be modeled as poly-purines. As, this could help the community in deciphering their identity.

Reviewer #3

(Remarks to the Author)

This is an important paper in the field and beyond, identifying a new role for AP2 domain containing proteins in *Toxoplasma* and other apicomplexans, and providing new structural information about how the mitoribosome works in these organisms despite highly reduced mito genome complexity and fragmentation. It is lovely work.

only one comment: to change the section title "The four mtAP2s are piecing together rRNA fragments in the mitoribosome" specifically the phrase "are piecing together" (perhaps replace with "assemble")

Version 1:

Reviewer comments:

Reviewer #1

(Remarks to the Author)

This reviewer appreciates the thoughtful response to the comments brought up in the first review. The manuscript has been greatly improved and is much easier to read and understand.

Reviewer #2

(Remarks to the Author)

The authors have addressed my concerns. I am happy with the manuscript in its current state.

However, I notice some issues with references.

line 404: ref 38 points to Waltz et al, Nat Commun 12, 7176 (2021) which does not seem to be the right one for ciliate but for green alga; ref 9 points to Nature Plants 6, 377-383 (2020) which is a report of cauliflower mitoribosome and thus does not seem to be apt for 'green alga'. These should be corrected.

In this context I further suggest that the following references be added:

lines 294, 404: <https://doi.org/10.7554/eLife.59264>. This is an important work that reports the structure of a ciliate mitoribosome and certainly relevant to be cited here.

line 404: <https://doi.org/10.1038/s41467-022-33582-5>. This work reports the mitoribosome structure from *P. magna* a green alga and seems to be quite relevant here.

With these corrections, I fully recommend the publication of this work.

Thank you

REVIEWER COMMENTS

Reviewer #1 (Remarks to the Author):

We wish to thank the reviewer both for their positive reaction to our manuscript as well as their constructive comments. We have given serious consideration to these comments and have made several editorial changes to address the manuscript complexity concerns raised.

This robust and extensive body of work describes the components and structure of the *Toxoplasma gondii* mitoribosome. The structures of this molecular machine are quite different, in comparison to what we know from model organisms, being composed of highly fragmented ribosomal RNAs. There is both high significance and innovation in the manuscript and it should be of impact beyond the field of parasitology. The work presented is of high quality and well executed. Nonetheless, the manuscript is very dense and difficult to follow given the large number of figure panels with little to no description. Streamlining what is presented in the main figures with a focus on the main storyline would aid in making the manuscript more accessible to the readers.

To address this, we have expanded the main text and the figure legends to provide clearer and more detailed descriptions of each panel.

Major

This manuscript is extremely dense in data. The authors should do everything possible to ease its interpretation and reading by simplifying and prioritizing what they include.

Following are a few items that could address this issue:

- Including data on *P. berghei* has limited impact, consider not including it or making *Plasmodium* data a stand-alone supplementary figure.

To address the issue of the manuscript being too dense in data, we have prioritized the data presented to enhance clarity and readability. Therefore, as suggested by the reviewer and recommended by the editors, we have kept the *Plasmodium berghei* data but moved them to a stand-alone supplementary figure.

- If *Plasmodium* data is kept, be clear in the text and figure legend about whether data is from *Toxoplasma* or *Plasmodium*.

We have thoroughly revised the text and figure legends to explicitly indicate whether the data pertains to *Toxoplasma* or *Plasmodium* in each instance.

- The experiment using the MPP knockdown (fig 1g and 1h) is clever but superfluous to the main story. Either take it out or make it a supplementary data. Same with data in 1i. This will unclutter Figure 1.

We agree that these data, while supporting the characterization of the mtAP2s, are not essential to the main story and slightly distracting. To streamline Figure 1 and improve the manuscript's focus, we have opted to remove the data related to MPP-alpha and targeting.

- The experiment using mtAP2-2 and 2-3 tagged with an AID (Auxin Inducible Degron) in Extended Fig1e is not necessary, and it is not mentioned in the text.

As recommended this control experiment has been removed.

- The experiment in extended data 2f with the *Eimeria* homolog is odd as that homolog had not been characterized in terms of function and localization within *Eimeria*. Seems more appropriate to use the *Plasmodium* version to keep a cohesive story in the paper. We appreciate the reviewer's feedback but have chosen to retain the *Eimeria* homolog experiment in Supplementary 3f, as it highlights evolutionary conservation across apicomplexan parasites. While the homolog has not been fully characterized, its inclusion. We have clarified in the text how trans-genera complementation of an AP2 domain broadens the perspective of our work.

There is a concerning omission of prior published work on the mitoribosome. In particular, the authors should incorporate prior work on their discussion and cite Shikha et al 2022 and Lacombe et al 2019.

We have cited and acknowledged the relevant findings from Shikha et al. (2022) and Lacombe et al. (2019) in the revised manuscript.

Minor

In the main figure 2 (a and b), the amount of data makes it difficult to interpret all the graphs. The colors used by the authors are difficult to distinguish on the screen. I strongly suggest that the authors modify the color scheme for both the proteomics and RNAseq data (Complex II and V, for example).

To enhance readability and interpretation, we have modified the color scheme for both the proteomics and RNA-seq data, particularly for Complex II and V. We trust the revised color palette to ensure that the data is easily distinguishable both on-screen and in print.

Figure 3 has the same problem, as the grey-filled dots are difficult to distinguish in the graphs.

Done as well.

Line 102, LOPIT is empirical data and not a prediction.

Corrected.

Line 120 indicate that the fitness data is for *Toxoplasma*

Done.

Line 122, for clarity perhaps indicates which ones were knockdown with the tet inducible system and which with the snRNP mediated system

Done as recommended.

In figure 4, the relative position of panel f is not ideal.

Position of panel f has been improved.

In the figure legend for figure 4, (e) appears to be missing after “ and the small ribosomal subunit side”.

The figure legend has been edited for completeness.

Reviewer #2 (Remarks to the Author):

Title: Apicomplexan Mitoribosome from Highly Fragmented rRNAs to a Functional Machine

Summary

Wang et al., report that four ApiAP2s (1-4) localize to the mitochondrion in human parasite *T. gondii*. They further establish that these transcription factor-like proteins are required for the structural integrity of the mitoribosome but not for transcription. With a combination of single particle cryo-EM, RNA sequencing and mass spectrometry, the authors determine the structure of the mitoribosome. This provides the structural basis not only for the role of AP2s in the mitoribosome structure but also how a highly fragmented rRNA (composed of over 56 fragments) forms the mitoribosome. Additionally, the authors also identify RNA-binding RAP and HPR proteins in the structure leading to insights as to how they stabilize the mitoribosome.

This study provides insights into the various structural mechanisms enabling the assembly of mitoribosome from numerous short rRNA fragments. This is an important contribution to our growing understanding of the drivers of mitoribosome evolution. Since, this comes from a human parasite, this is also very useful for future rational drug development efforts.

The quality of the data is good and the manuscript reads well. The illustrations are clear and unambiguous.

I am happy to have reviewed this manuscript and recommend the publication of this work. However, I would like to make the following recommendations towards the cryo-EM data interpretation and discussion.

We thank the Reviewer for generously reviewing our models, analyzing them thoroughly, and providing valuable suggestions to enhance their quality. To address their points, we have made several modifications, which are detailed below. We hope that they will find the revised manuscript significantly improved and now suitable for publication.

As I understand, the number of fragments are determined to be 56 (33 in LSU and 23 in SSU) by a combination of cryo-EM and RNA-seq. However, these include 21 chains which are modeled as ulr/usr as the sequences could not be identified from RNA-seq data. Since, the identity of these have not been established, it cannot be ruled out that some of these are (unmodeled) extensions of the identified rRNA chains. Therefore, I think the number 56 could be an overestimation and the authors could consider rephrasing 'over 56 fragments' in the abstract.

There are 25 fragments in the LSU and 15 in the SSU that are assigned, and 23 fragments that are not assigned across both subunits (totaling 63 fragments). Our RIP-

Seq data identified a total of 43 rRNA fragments, two of which were not detected in the structural analysis. We agreed with the Reviewer that some of those unassigned leftover rRNA (ulr) pieces are most likely extensions of the identified rRNA chains or only parts of the same rRNA piece. To address this concern, we will revise the phrasing in the abstract from “over 56 fragments” to “over 40 fragments” to better reflect this uncertainty.

The validation report shows sequence discrepancy in the chain HJ which should be addressed.

The reviewer identified a discrepancy in chain hJ of the SSU head model, where the cross-reference was incorrectly annotated during deposition. The reference code has been updated from A0A7J6KBH4 to S8F5G8.

Lines: 283, 284: “We chose to represent unassigned fragments as poly-U fragments in our deposited structures and to give them generic names based on their provenance “usr” and “ulr” for unassigned SSU and LSU rRNAs, respectively.”

This sentence has been replaced by: “We chose to represent unassigned fragments as poly-U fragments in our deposited structures and to label them with the generic name “ulr”.

However, in the PDB validation reports, I could only find reference to ‘ulr’ chains. Kindly check.

The text has been modified accordingly.

The values of model vs data cross correlation for LSU reported in extended table 1 seem way too low, probably a typo?

We thank the Reviewer for noticing this point. The CC values for the LSU have been corrected in Table 1. The updated values are, CC_(mask): 0.6036, CC_(box): 0.6089, CC_(peaks): 0.6029, and CC (volume): 0.6720.

It would be great to see a brief comparison between rRNA fragmentation patterns determined from the current work and those published recently, from *Polytomella magna* (10.1038/s41467-022-33582-5) and *Chlamydomonas reinhardtii* (10.1038/s41467-021-27200-z). Panels could be added to Extended data fig. 9.

The reviewer suggested including a comparison of fragmentation patterns between our work and recent publications. While we appreciate this idea, this point is beyond the scope of this study, and we have decided to reserve it for future work, as adding secondary analyses at this stage could overcomplicate an already dense manuscript.

The following suggestions are directed at the mitoribosome model:

In SSU head, there is a large unmodeled density likely an rRNA helix continuous with chain hG (resi 40-41;). The following resi 44-56 of hG (terminal residue in the chain) form a helix with chain bN (SSUB) resi 74-82 , but, do not show the usual Watson-Crick base-pairing with the corresponding residues. This indicates that chain hG is likely not correctly modeled and can be improved so as to accommodate more residues (likely as

a helix) in the unmodeled density, using WC-pairing within chain hG and with resi 74-82 of chain bN as a guide.

We have rebuilt chain hG starting from residue 56. This reconstruction has resulted in a more accurate Watson-Crick pairing and the identification of a new loop that was previously unmodeled. We hope that these new data address satisfactorily the concern raised.

A large unassigned volume in SSU head, that runs along chain HK (resi 41-65) and continues to fill up the inter-facial cavity between chains HF, HK, HO and HP. This is likely an RNA chain. The authors could perhaps assign this as a Poly-U chain if sequence can not be determined.

We have reviewed the map for any unassigned volumes, which led to the identification of a new poly-U RNA chain, designated as chain hS, placed in the volume specified by the reviewer.

In SSU head, two large unmodeled densities (very likely protein) closely bound to chain HR.

A) One looks like a helix-loop-helix likely to be the missing residues from chain B4 (resi 43-94).

B) The other could be a part of chain Yq. However, it seems to be weakly connected (at lower threshold) with another unmodeled density sandwiched between chains Br and Yi.

Authors could consider modeling these as poly-Ala if the sequence identity could not be determined as the backbone can still probably be traced.

Residues 43-94 of chain B4 were not modeled due to poor resolution in that region, which hindered our ability to infer internal residue contacts.

At SSU-LSU interface, there is an unmodeled density that connects RNA chain bS (terminal) to RNA chain bE (at residue 94-95). Following resi 95 of chain bE, the sequence does not appear to match the density very well. For example, 1) resi 96-99 are completely outside the density; 2) resi 103 and 104 are modeled as U and C, respectively, however, the density suggests that these should be purines.

Residues 95-104 of chain bE were changed to poly-U and connected to chain bS, with coordinates adjusted for a better fit to the map.

In the LSU, at the interface between chains L4 and Lh there is a density potentially corresponding to 6-7 nucleotides. At this resolution it looks like a poly-purine stretch. Additionally, chains, IU (LSU) and bY (SSU) built as poly-uridine look (from the density) almost entirely like long poly-purine chains. They could be considered to be modeled as poly-purines. As, this could help the community in deciphering their identity.

A new poly-U chain, designated as IY, has been placed at the interface between chains L4 and Lh.

We appreciate the reviewer's suggestion to change specific chains from poly-U to poly-R. However, we believe that maintaining poly-U as the standard for unassigned rRNAs will reduce the potential for misinterpretation within the community. We hope that the Reviewer will agree with our reasoning.

Reviewer #3 (Remarks to the Author):

This is an important paper in the field and beyond, identifying a new role for AP2 domain containing proteins in *Toxoplasma* and other apicomplexans, and providing new structural information about how the mitoribosome works in these organisms despite highly reduced mito genome complexity and fragmentation. It is lovely work.

only one comment: to change the section title "The four mtAP2s are piecing together rRNA fragments in the mitoribosome" specifically the phrase "are piecing together" (perhaps replace with "assemble")

We appreciate the reviewer's support to the work. We have revised the section title that reads now "The four mtAP2s assemble rRNA fragments in the mitoribosome."

REVIEWERS' COMMENTS

Reviewer #1 (Remarks to the Author):

This reviewer appreciates the thoughtful response to the comments brought up in the first review. The manuscript has been greatly improved and is much easier to read and understand.

We appreciate the reviewer's support to the work.

Reviewer #2 (Remarks to the Author):

The authors have addressed my concerns. I am happy with the manuscript in its current state.

However, I notice some issues with references.

line 404: ref 38 points to Waltz et al, Nat Commun 12, 7176 (2021) which does not seem to be the right one for ciliate but for green alga; ref 9 points to Nature Plants 6, 377-383 (2020) which is a report of cauliflower mitoribosome and thus does not seem to be apt for 'green alga'. These should be corrected.

Corrected.

In this context I further suggest that the following references be added:

lines 294, 404: <https://doi.org/10.7554/eLife.59264>. This is an important work that reports the structure of a ciliate mitoribosome and certainly relevant to be cited here.

line 404: <https://doi.org/10.1038/s41467-022-33582-5>. This work reports the mitoribosome structure from *P. magna* a green alga and seems to be quite relevant here.

We thank the reviewer for highlighting this point. The two highly related references have been cited in the revised manuscript.